# Orbitrap noise structure and method for noise unbiased multivariate analysis

Michael R. Keenan [1,9], Gustavo F. Trindade [2,9], Alexander Pirkl[3], Clare L. Newell [4], Yuhong Jin [4], Konstantin Aizikov[5], Andreas Dannhorn [6], Junting Zhang[2], Lidija Matjačić[2], Henrik Arlinghaus [3], Anya Eyres [2], Rasmus Havelund[2], Richard J. A. Goodwin [6], Zoltan Takats[7], Josephine Bunch[2], Alex P. Gould [4], Alexander Makarov [5,8] & Ian S. Gilmore [2] ✉

Orbitrap mass spectrometry is widely used in the life-sciences. However, like all mass spectrometers, non-uniform (heteroscedastic) noise introduces bias in multivariate analysis complicating data interpretation. Here, we study the noise structure of an Orbitrap mass analyser integrated into a secondary ion mass spectrometer (OrbiSIMS). Using a stable primary ion beam to provide a well-controlled source of ions from a silver sample, we find that noise has three characteristic regimes: at low signals the Orbitrap detector noise and a censoring algorithm dominates; at intermediate signals counting noise specific to the ion emission process is most significant; and at high signals additional sources of measurement variation become important. Using this understanding, we developed a generative model for Orbitrap data that accounts for the noise distribution and introduce a scaling method, termed WSoR, to reduce the effects of noise bias in multivariate analysis. We compare WSoR performance with no-scaling and existing scaling methods for three biological imaging data sets including drosophila central nervous system, mouse testis and a desorption electrospray ionisation (DESI) image of a rat liver. WSoR consistently performed best at discriminating chemical information from noise. The performance of the other methods varied on a case-by-case basis, complicating the analysis.

A study of instrument noise may, at first, seem the esoteric preserve of the specialist but a better understanding can make a profound difference to the scientific value of data. Mass spectrometry (MS) is one of the most widely used analytical techniques but there are relatively few noise studies. This is surprising since the noise in MS is typically heteroscedastic (noise level varies with peak intensity) which complicates computational methods to interpret data such as multivariate and machine learning-based approaches. For example, the issue has been demonstrated for principal component analysis (PCA) of secondary ion mass spectrometry (SIMS) data acquired with a time-of-flight (ToF) mass spectrometer under conditions that the secondary ions are Poisson-distributed[1]. In that case, the variance is equal to the peak intensity so that the first principal component is typically dominated by intense peaks whilst low intensity peaks, often the most analytically

[1]Independent, Georgetown, TX, USA. [2]National Physical Laboratory, NiCE-MSI, Teddington, UK. [3]IONTOF GmbH, Münster, Germany. [4]The Francis Crick Institute, London, UK. [5]Thermo Fisher Scientific, Bremen, Germany. [6]AstraZeneca, Cambridge, UK. [7]Imperial College London, London, UK. [8]Biomolecular Mass Spectrometry and Proteomics, Bijvoet Center for Biomolecular Research and Utrecht Institute for Pharmaceutical Sciences, University of Utrecht, Utrecht, The Netherlands. [9]These authors contributed equally: Michael R. Keenan, Gustavo F. Trindade. ✉e-mail: ian.gilmore@npl.co.uk

important, are buried in higher order principal components that capture mainly noise. A method to rescale the data was developed so that these low intensity peaks with chemically induced variance were captured in the leading principal components[1,2]. Since the use of computational methods in mass spectrometry[3,4] is rising rapidly the importance of correct statistical treatment of noise will grow.

The Orbitrap™ mass analyser[5] is a widely used ion trap mass spectrometer with a compact design that achieves high mass resolving power and high mass accuracy. Indeed, there are now many thousands of Orbitrap instruments used in all manner of analytical measurements[6]. The Orbitrap[5] MS has been extensively described elsewhere[7–9]. Briefly, ions from a (quasi-)continuous source are accumulated in a curved ion trap known as the C-trap. Ions are collected for a period of time after which they are injected in a short pulse into the trap with hyper-logarithmic potential during its ramping towards the final voltage. The trapped ions oscillate along the major axis of the central electrode with a frequency inversely proportional to the square root of their mass-to-charge ratios. An image current is created on the pair of outer electrodes that is measured with time using a differential amplifier. The time-domain transient signal is digitised and can be converted to the frequency (and hence mass) domain by a Fourier transform (FT) or its derivatives (eFT[10], aFT[11]), as well as non-FT approaches[12–17].

The detector noise in Fourier transform ion cyclotron resonance (FT-ICR) mass spectrometers has been studied in detail[18]. These spectrometers, first introduced in 1974[19], use a magnetic rather than electrostatic field to trap the ions but share the method for detection of ions using sensitive measurement of the image charge. Consequently, the development of Orbitrap technology could use much of this knowledge. Three main types of noise in FT-ICR have been identified[18]: (1) source-limited noise, (2) detector-limited noise and (3) fluctuation noise. Source-limited noise relates to the random ion generation process, typically described by a Poisson distribution. Detector-limited noise is independent of the signal strength and is dominated by thermal noise in the preamplifiers. Fluctuation noise has power (squared magnitude) inversely proportional to frequency. A comprehensive study in 1993 of the magnitude spectrum peak intensities of an FT-ICR instrument found the noise was described by a Rayleigh distribution, as expected from the Fourier transform of uniform Gaussian noise in the time domain, with a noise level equivalent to 59 ions[20]. A detection limit of 177 ions was determined for a signal-to-noise ratio of 3:1. Thermal noise from the image current pre-amplifier has also been found to be the main source of noise in an Orbitrap signal[21–23].

Recently, we introduced OrbiSIMS[24] using an instrument where a primary ion beam with stable beam current impacts a sample to generate secondary ions which are accelerated by an extraction electrode and can either pass directly to a ToF MS or be deflected to a transfer system that sends them to the Orbitrap MS (Fig. 1a). The high stability of the source provides an excellent platform to study noise in the Orbitrap MS and the complete OrbiSIMS instrument.

Here, we report the development of a probabilistic generative model for Orbitrap MS data that fully accounts for the distribution and second-order statistics of data acquired from a pure silver test sample over five orders of magnitude in signal intensity. The model emulates the discrete nature of ions and is of sufficient generality to be applied to Orbitrap data more broadly. The model enables questions to be addressed, such as the detection limit, directly in terms of numbers of ions. It also brings insights into noise structure that improves treatment of heteroscedastic data and reduce undue influence of noise (noise bias) in multivariate statistical methods.

## Time domain signal

Broadly speaking, data production in OrbiSIMS involves two processes: (1) generating a mixture of secondary ions and transferring them to the Orbitrap mass spectrometer; and (2) Orbitrap ion

detection by image-current measurement and data processing. An important aspect of data processing is that low-intensity signals are heavily censored to manage the volume of data stored. Data elements below an instrument-determined noise threshold are set to zero. Given $n_i$ ions of type $i$ oscillating with axial amplitude $\Delta z$ at angular frequency $\omega_i$, the fundamental equation describing the time course of the Orbitrap image current is[5]:

$$I(t|n_i) = -q_i n_i \omega_i \frac{\Delta z}{g} \sin(\omega_i t) \tag{1}$$

where $q_i$ is the charge on the ion and $g$ is the "effective gap" between detection electrodes, which may depend on $i$. This current is detected by a preamplifier with impedance ($Z_i$) dominated by the combined capacitance $C$ of detection electrodes and input preamplifier transistors: $Z_i \approx 1/(\omega_i C)$, therefore its output $V(t|n_i) = I(t|n_i)*Z_i$ becomes largely independent on frequency and will be treated as a constant here. For simplicity, detector geometry and ion-dependent quantities can be combined into a single scale factor relating voltage to the number of trapped ions:

$$V(t|n_i) = \sqrt{2}An_i \sin(\omega_i t) \tag{2}$$

$An_i$ is the root-mean-square amplitude of the sinusoidal component of ions oscillating at frequency $\omega_i$. The frequency-domain magnitude spectrum $S$ is obtained via Fourier transform and a conversion to a mass spectrum is accomplished according to the inverse proportionality between frequency and the square root of the mass-to-charge ratio ($m/q$):

$$\omega = \sqrt{\frac{\alpha}{(m/q)}} \tag{3}$$

where $\alpha$ is a proportionality constant. The observed mass spectrum is the sum of signal and noise: $X = S + N$. The variance due to noise at frequency $\omega$ in the magnitude spectrum is the sum of the three contributions[18]:

$$\sigma^2(\omega) = \sigma_S^2(\omega) + \sigma_W^2 + \sigma_F^2(\omega) \tag{4}$$

$\sigma_S^2(\omega)$ is the variance of source-limited noise in the signal. This is shot noise that originates from the discrete nature of the ions and has a standard deviation that varies with the square root of $S$. Detector-limited noise, denoted $\sigma_W^2$, represents additive white Gaussian noise (AWGN) in the time-domain signal. $\sigma_F^2(\omega)$ is the variance of fluctuation noise. The power spectrum of fluctuation noise, also known as $1/f$ or flicker noise[25], varies inversely with frequency. $1/f$ noise is the dominant source of noise at very low frequencies (high mass) with few ions oscillating. AWGN is the largest contributor to the spectral background in the present measurements.

## Statistical properties of Orbitrap signals and noise

The statistical distribution of data from an FT-based acquisition has been considered in the context of MRI imaging[26] and it accommodates the full range of S/N. For a constant signal magnitude $S$ and time-domain noise standard deviation σ, the data are found to follow the Rician distribution[27]. The Rician distribution would be appropriate for mass spectral data if the number of ions, $n_i$ in Eq. (2), is constant with $s_i = An_i$. However, $n_i$ is not fixed, in general, but rather is randomly drawn from some discrete distribution with probability mass function $D$. In that case, the distribution of the $i^{th}$ mass peak height is given by a weighted sum of Rician distributions with weights given by the distribution of ions

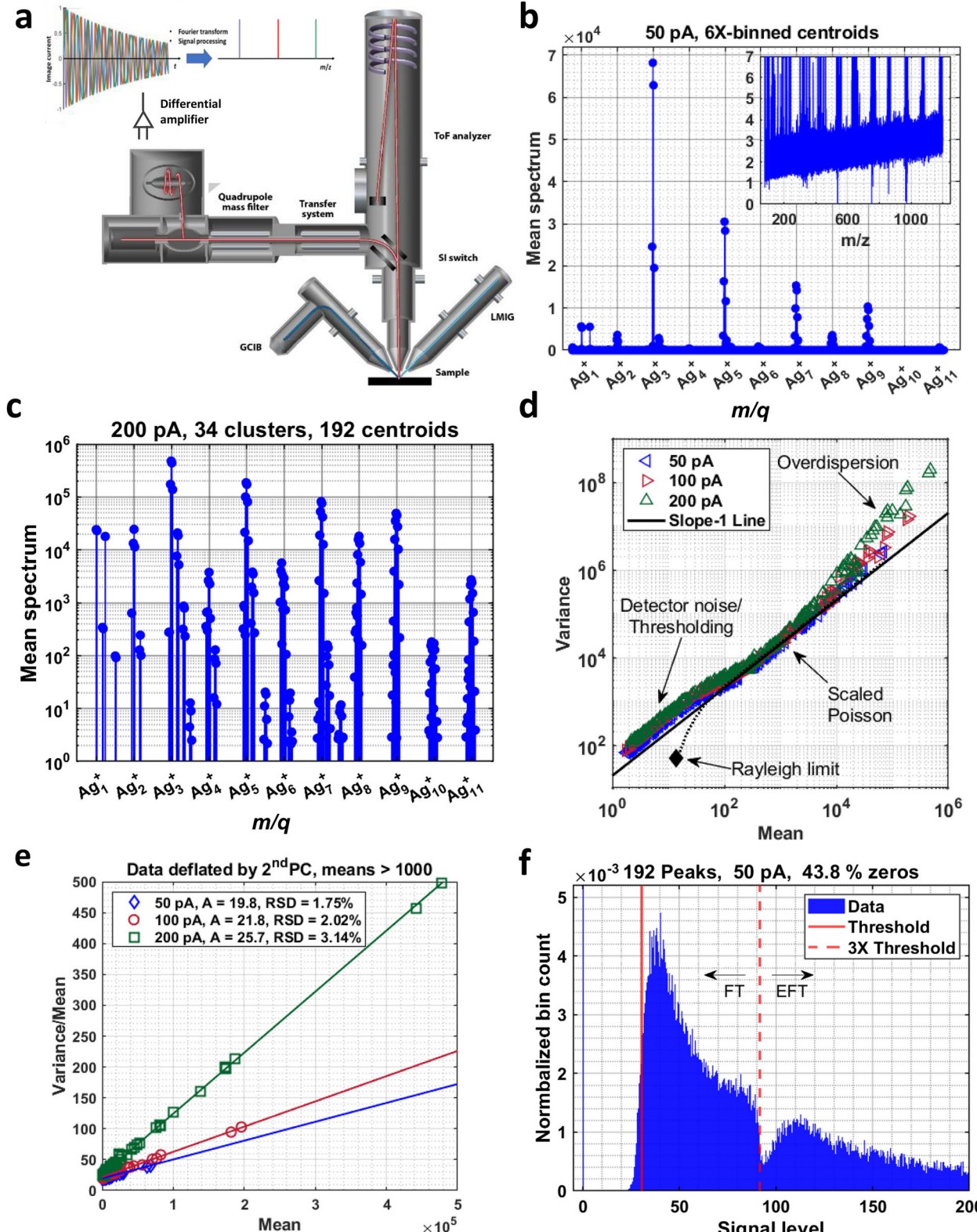

**Fig. 1 | Measurement of noise in an Orbitrap mass spectrometer. a** Schematic of an OrbiSIMS instrument (adapted from Ref. 24 with permission from Springer Nature BV, Copyright 2017). **b** Centroids mean spectrum of a 1000-depth profile of silver acquired at a beam current of 50 pA. The inset shows a slight increase in the background noise with increasing mass. **c** Log-scale mean spectrum of 192 integrated peak windows showing more than 5 orders of magnitude variation in peak intensity. **d** Relationship between variance and mean for 192 peaks at each of three beam currents. If the depth-to-depth variation in the number of trapped ions is Poisson-distributed, the points should fall on a line having a slope of one. **e** Variance-to-mean vs mean relationship for all datasets in linear scale, showing fitted scaling factor A and overdispersion RSD. **f** Histogram of all silver-peak intensities for the 50-pA dataset at low signal levels. The threshold is evident and about 43.8% of the data entries are represented in the (truncated) zero peak. The Enhanced Fourier Transform is applied once a peak exceeds 3 times the threshold.

contributing to that peak as shown in the first of Eq. (5).

$$f_S(s_i, \sigma) = \sum_{j=0}^{\infty} D(j) \times Rician(Aj, \sigma)$$

$$f(s_i) = \delta(s_i) \int_{-\infty}^{K\sigma} f_S(s_i, \sigma) ds_i + f_S(s_i, \sigma) H(s_i - K\sigma) \quad (5)$$

The second of Eq. (5) accounts for data thresholding where it is assumed that all signal-magnitude values less than a multiple $K\sigma$ of the time-domain noise standard deviation are set to zero. Here $\delta$ is the Dirac delta function and $H$ is the Heaviside step function. We denote this generalized probability function by a weighted sum of Rician (WSoR) distributions. It is worthwhile to note that at S/N > 3, the Rician distribution is well-approximated by a Gaussian. As sums of Gaussians are themselves Gaussian, the sum of Ricians will appear Gaussian at high signal levels

## Statistics of OrbiSIMS ion generation and transfer

OrbiSIMS employs a primary ion beam to generate a stream of secondary ions whose distribution reflects the composition of the sampled material. To produce a single mass spectrum, ions are generated at a rate $\lambda$ for time $t_c$ and transferred to the Orbitrap analyser with efficiency $\tau$. Secondary ion generation is a Poisson process. If $\lambda$ is strictly constant, $N_0$, the total number of ions generated given $\lambda t_c$, is itself Poisson-distributed: $N_0|\lambda t_c \sim Poisson(\lambda t_c)$ and the total number of ions trapped $N = \tau \lambda t_c$. However, if the trapping rate is subject to variation or noise, this process can be generalized. Allowing $\tau \lambda t_c$ to vary according to some distribution with mean $\mu_N$ and variance $\sigma_N^2$, the unconditional distribution of $N$ is characterized by:

$$E[N] = E[E[N|\tau\lambda t_c]] = E[\tau\lambda t_c] = \mu_N$$
$$Var(N) = E[Var(N|\tau\lambda t_c)] + Var(E[N|\tau\lambda t_c]) = \mu_N + \sigma_N^2 \quad (6)$$

The variance of a Poisson random variable is equal to its mean. By allowing the ion trapping rate to vary, the variance of $N$ increases by the variance of $\tau \lambda t_c$ and becomes over-dispersed with respect to Poisson. It will be convenient to rewrite the variance:

$$Var(N) = \mu_N + \left(\frac{\sigma_N^2}{\mu_N^2}\right)\mu_N^2 = \mu_N + R_N^2 \mu_N^2 \quad (7)$$

Here, $R_N$ is the relative standard deviation RSD attributed to overdispersion. Over-dispersed count data are commonly modelled as negative binomials.

## Second-order statistics of a mass spectrum

The Orbitrap mass analyzer partitions a mixture of $N$ trapped ions into $k$ bins defined by the frequency resolution. The distributions of the mass-separated ions are conveniently modelled by the multinomial distribution and represented by vector **n**. We next convert the number of ions to a mass spectrum **s** via multiplication by the ions-to-signal scale factor: **s** = $A$**n**. For simple scaling, the expected statistics of the mass spectral signal are:

$$E[\mathbf{s}] = A E[\mathbf{n}] = \boldsymbol{\mu}$$
$$Var(\mathbf{s}) = A^2 Var(\mathbf{n}) = A E[\mathbf{s}] + R_N^2 E[\mathbf{s}]^2 \quad (8)$$
$$Cov(\mathbf{s}) = diag(A E[\mathbf{s}]) + R_N^2 E[\mathbf{s}]E[\mathbf{s}]^T$$

Here, Cov(**s**) is the full covariance matrix. These expressions account for the source-dependent variations of OrbiSIMS signal in the absence of detector noise. The variance is a quadratic function of the mean, and the off-diagonal elements of the covariance matrix comprise the outer product of the mean spectrum scaled by the square of the RSD due to overdispersion.

Taken together, Eqs. (5)–(8) constitute a comprehensive model for OrbiSIMS data generation and noise that is appropriate to the full range of S/N (details provided in Supplementary Note 1). In addition to an instrument-specific thresholding parameter, the noise distribution is characterized in terms of four parameters: $A$ and $R_N$ that relate to ion counting statistics, and the intrinsic detector noise variances $\sigma_W^2$ and $\sigma_F^2$. These need to be estimated from experimental data.

## Results

### Description of the measured signal

Measuring the noise distribution in the Orbitrap spectra requires a source of ions that is stable over the time frame of the measurements. We used the bismuth liquid metal ion gun in the OrbiSIMS, which has a current stability <1% RSD over a 10-minute period. The beam current can be readily changed from 0.5 pA to 1000 pA, giving fine control over the number of secondary ions generated. 1000 nominal replications of mass spectra of a metallic silver sample were acquired over the course of a depth profile at each of three secondary-ion generation rates using ion beam currents of 50 pA, 100 pA and 200 pA (Supplementary Note 2). These cover the range from normal operation through to saturation at 200 pA beam current. At 200 pA, the $Ag_3^+$ intensity approaches $10^6$ arb. u. which leads to non-linearity effects for higher mass ions (Supplementary Note 3). These effects are outside the normal recommended Orbitrap operation but are progressive and can be encountered for example, in OrbiSIMS analysis of organic semiconductors[28]. In depth profiling experiments, the problem of saturation can be mitigated by using an Adaptive Ion Injection System (Supplementary Note 1).

Each spectrum was produced by sampling the time-domain transient at 4.096 MHz for 0.512 s. After triple-zero padding and computing the single-sided magnitude, each spectrum comprises approximately 2.5 million points evenly spaced in frequency and spanning the mass range from 80 to 1200 Da. As the vast majority of these mass channels contain only noise, only data exceeding a threshold value are recorded in the RAW data file.

Orbitrap data can be accessed in several formats. These include spectral-peak profiles for those peaks retained in the dataset, full-resolution peak centroids, and peak lists (OrbiSIMS). A description and comparison of these data formats is detailed in Supplementary Note 4. For reasons outlined there, we have chosen to analyse data in the full resolution centroid format. In the present case, this format provides a wealth of data for assessing the background noise and can be used to construct peak lists for specific ions. In the latter case, since the location of the centroid from a given peak may vary from scan to scan, we obtain peaks by summing centroids within 5-point intervals centred on the local maxima of the mean spectrum, which guarantees we collect the right channel for a given scan. The inset in Fig. 1b shows that the background signal increases slightly with mass due to fluctuation noise. The mean signal levels shown are artificially low since heavy censoring of low-intensity signals forces most of the data elements in any pure-noise mass channel to be zero.

### Estimation of source-limited noise

To characterise the statistics of mass peaks attributed to trapped ions, we constructed datasets based on a peak list consisting of 192 peaks in 34 chemically distinct clusters. 132 of the mass peaks correspond to isotopes of the clusters $Ag_x^+$, $x = 1 \ldots 11$, and $BiAg_y^+$, $y = 0 \ldots 9$. The remaining clusters contain Ag with either $H_2O$ or additives such as Mn[29]. Notably, a peak corresponding to $Cs^+$ (surface contamination) is present. Unlike the other peaks, $Cs^+$ exhibits depth-dependent intensity variation. This provides an additional source of variation that should be accounted for in a factor model of the silver data. A log-scale plot of the mean peak spectrum of the 200-pA dataset (Fig. 1c) shows more than 5 orders of magnitude variation in peak intensity.

The relationships between peak means and variances are shown in Fig. 1d for the three datasets. According to Eq. (8), Poisson-like variance is directly proportional to the mean. On a log-scale plot, this implies that the slope will be one and the intercept will be the log of the scale factor $A$. At low signal levels, detector noise dominates and, in the absence of censoring, the variance and mean should approach constant values reflecting the S = 0 limit of the Rician distribution (i.e., Rayleigh distribution) (Fig. 1d). Whilst there is a range of moderate signal levels that conform to the model proposed by Marshall and Verdun[18], the data at both high and low intensities deviate significantly. At low intensity, setting some data elements to zero reduces the mean and inflates the variance. The data at high intensity are over-dispersed (*"over-dispersion"* in Fig. 1d) with respect to the scaled-Poisson expectation with the extent of over-dispersion increasing with both signal intensity and ion generation rate.

At high signal levels where detector noise is negligible, Eq. (8) predicts that the variance of a mass peak should be a quadratic function of its mean. The linear term describes counting noise that varies independently from ion to ion (additive noise) and the quadratic term accounts for variation in the total number of ions in the trap that affects the signal level of all types of ions in the same way (multiplicative noise). This variance curve can be directly fit to a quadratic of the form Eq. (8) to yield estimates of $A$ and $R_N$. However, our preferred method is to fit a linear expression. Dividing the variance vector by the spectral mean:

$$\frac{\text{Var}(\mathbf{s})}{E[\mathbf{s}]} = A + R_N^2 E[\mathbf{s}] \qquad (9)$$

Division is performed elementwise, and the expectations are estimated as the sample mean and variance of the data matrix. Equation (9) suggests that a plot of variance-to-mean *vs* mean should be linear with intercept and slope given by the ion-count-to-signal scale factor $A$ and squared relative standard deviation due to overdispersion $R_N^2$, respectively. Such plots for the three ion currents are shown in Fig. 1e. Here, some minor instrumental artefacts have been removed from the data as detailed in Supplementary Note 3. Both the scale factor and level of overdispersion increase with beam current. The intercepts of the regression lines are fixed by the scale factor $A$. The fact that the noise ratios for all high-intensity peaks of a given type fall on the same line reinforces the notion that within the sensitivity of our experiment, the scale factor is independent of mass and chemical identity. Finally, it is noted that for the 200 pA dataset, which is most highly over-dispersed, the linear and quadratic fits agree within statistical uncertainty.

### Estimation of detector-limited and flicker noise

At low intensities, the data points are parallel to the slope-1 line, suggesting a Poisson-like process governs the relationship (Fig. 1d). However, this behaviour does not originate with ion counting as in the high intensities. At low intensities, Poisson-like behaviour reflects not random variation in the number of trapped ions but rather random variation in the number of peaks that exceed a zeroing threshold. The Orbitrap threshold appears to be set at a fixed multiple of a locally determined random-noise standard deviation. This approach is called the "N-sigma" rule[30]. The so-called reduced profile mode yields highly sparse data as the signal intensity approaches the noise level. In fact, given replicate measurements, the sparsity can be used to filter out noise channels in the mass spectrum[31]. The threshold is plainly evident and, overall, about 44% of all entries in the dataset have been zeroed (Fig. 1f). In "peaks" that are mostly noise, it is rare that the peak will exceed the threshold.

A second complication for interpreting low-signal noise has to do with the enhanced FT (eFT) spectral processing algorithm[10]. eFT calculates a weighted sum of magnitude and absorption spectra. Lower intensity parts of the spectra are represented in magnitude mode, but once the peaks exceed three times the threshold, absorption mode dominates. eFT is used to improve resolving power at half-height and includes filtering operations that can suppress some off-peak intensities. Figure 1f shows the distribution of low-intensity data elements for the 50 pA dataset. The transition from magnitude mode is responsible for the notch at about three times the threshold. For characterising background noise, however, eFT is irrelevant. Assuming 1% of noise "peaks" exceed the threshold, the probability that any noise peaks exceed the eFT onset is vanishingly small.

According to the general frequency-dependent noise model of Eq. (4), the total noise variance in the $i^{\text{th}}$ (signal-free) $m/q$ channel is the sum of white and $1/f$ components. The power, or equivalently variance, of $1/f$ noise is inversely proportional to frequency, which, in turn, is inversely proportional to the square root of $m/q$ according to Eq. (3). Hence, the variance due to $1/f$ noise at frequency $\omega_i$ is:

$$Power \propto \frac{1}{\omega_i} = \frac{\sigma_{F,i}^2}{\omega_i} = \sigma_F^2 \sqrt{m_i/q} \qquad (10)$$

where the proportionality constants have been combined into the parameter $\sigma_F^2$. Then the detector-related noise variance at frequency $\omega_i$ can be expressed:

$$\sigma_i^2 = \sigma_W^2 + \sigma_F^2 \sqrt{m_i/q} \qquad (11)$$

Given a series of signal-free mass "peaks", a plot of variance *vs* the square root of mass should be linear with the intercept and slope providing estimates of noise parameters $\sigma_W^2$ and $\sigma_F^2$, respectively.

Estimating detector-induced noise variance in Orbitrap data requires inferring characteristics of data that are largely excluded from the dataset. Noise in the time domain is taken to be uncorrelated AWGN that follows the Rayleigh distribution in the corresponding magnitude spectrum. Processing the time-domain transient in practice involves apodization with a Hanning window to minimize spectral leakage arising from the finite duration of the transient and triple-zero padding interpolation to improve the peak characterization[10]. Whilst noise in the magnitude spectrum remains Rayleigh-distributed, it becomes correlated in frequency/mass. That is, if a given data point exceeds the threshold, it is quite likely that adjacent points will exceed the threshold, as well. The patent literature[32] suggests that a peak will not be identified unless three consecutive points exceed the threshold. This literature also indicates that peak profiles contain eight points, which is consistent with our measured data (Fig. 2a). Note that most of those points lie below the threshold. In addition, apodization and zero-padding should result in a peak full width at half maximum of nine points for a pure sinusoid, which is significantly more than is observed. These observations show that peak profiles do not provide random samples of noise, rendering them unsuitable for our analysis. We therefore exclusively use centroids.

In terms of centroids of the measured silver data, approximately 1% of the data elements in a pure-noise channel exceed the threshold or are "detected". Given that detecting a non-zero centroid in a series of replicate measurements is a rare event, we might expect the distribution of the number of detects to be Poisson. The distribution of the number of detects in 1000 depths for all mass channels in the 50 pA dataset is in good agreement with the Poisson distribution having the same mean (Fig. 2b). In this model, which is again scaled Poisson, each detection should increment the mean by an amount that is, on average, the mean of all noise data exceeding the threshold, and the variance should increase in proportion to the mean with slopes close to one in accordance with the scaled Poisson assumption. Figure 2c shows the variance vs mean relationship of the low-intensity peaks for the three beam currents. The intercepts of the lines provide an estimate of how much each detect increments the mean. This

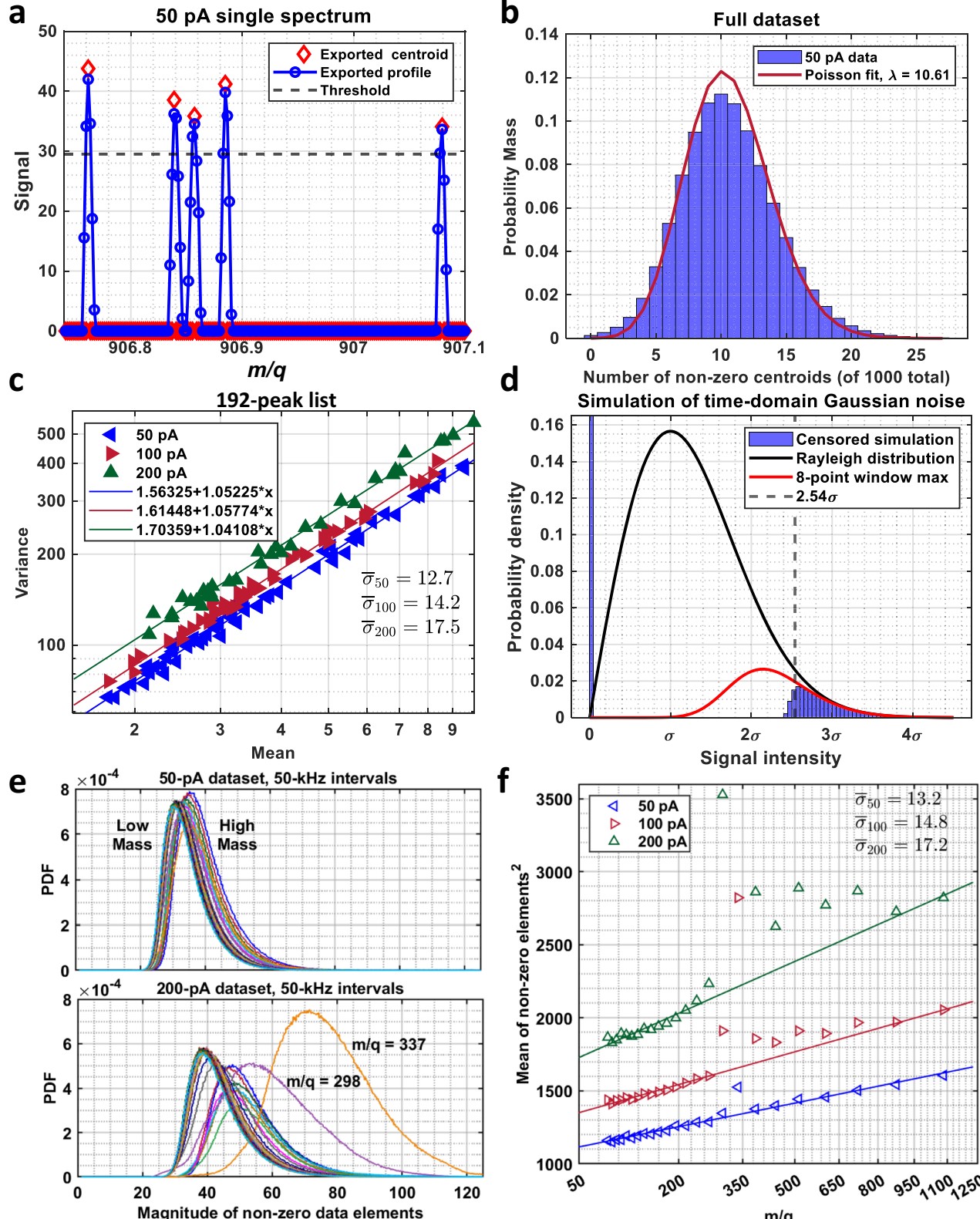

**Fig. 2 | Determination of model parameters. a** Spectral intensities in pure-noise channels near *m/q* 907 illustrate sparsity of censored data. **b** Distribution of non-zero centroids across the whole spectrum compared with a Poisson distribution. **c** Variance and mean relationship for very low intensities. **d** Comparison of a simulated distribution of censored centroid data with the tails of the corresponding uncensored Rayleigh and hypothesized centroid distributions. **e** Distributions of the 50 and 200 pA data within 50-kHz windows in the low-signal regime and **f** Squared means of the distributions shown in panel e plotted against the square root of m/q with line fits to estimate the detector-noise variance components. The integral average of the noise is indicated in the figure.

**Table 1 | All estimated noise parameters for the data acquired with different primary ion beam currents**

| Primary ion beam current (pA) | $A$ | $R_N$ | $\sigma_W$ | $\sigma_F$ | $\bar{\sigma}$ | $A / \bar{\sigma}$ | Excluded $m/q$ |
|---|---|---|---|---|---|---|---|
| 50 | 19.79(27) | 1.747(40) | 10.81(4) | 1.53(2) | 13.2 | 1.49 | 317 <$m/q$< 359 |
| 100 | 21.78(22) | 2.021(12) | 11.77(4) | 1.81(2) | 14.8 | 1.47 | 281<$m/q$< 792 |
| 200 | 25.72(36) | 3.143(59) | 13.07(7) | 2.27(3) | 17.2 | 1.49 | 251<$m/q$< 792 |

Source-limited parameters ($A$ and $R_N$) and detector noise parameters ($\sigma_W$, $\sigma_F$, $\bar{\sigma}$).

estimate, which does not take mass-dependence into account, provides an average value over the entire mass range.

There are two further tasks: relating the size of the increment to the underlying noise variance and making estimates of the variance as a function of mass. We assume that uncensored noise follows a distribution that is described by a single parameter so that varying the parameter simply rescales the plot. The Rayleigh distribution is shown with a second distribution[33] (Fig. 2d), which describes the expected maximum of Rayleigh realizations in an 8-point window. This distribution approximates centroids that are very unlikely to be spaced at less than 8-point intervals. The key observation is that the tails of the two distributions coincide at large signal values. Data that exceed the threshold are indicated by the shaded area. Here, the data were produced by simulating AWGN, applying Hanning-window apodization and triple-zero padding, and censoring the resulting magnitude spectrum.

Based on the simulation, the threshold was set at the $2.54\sigma$ point of the Rayleigh distribution to approximately match the sparsity of the simulated data to that of the measurements. The increment is then taken to be:

$$E[X|X > 2.54\sigma] = \frac{\int_{2.54\sigma}^{\infty} x \frac{x}{\sigma^2} \exp\left(\frac{-x^2}{2\sigma^2}\right) dx}{\int_{2.54\sigma}^{\infty} \frac{x}{\sigma^2} \exp\left(\frac{-x^2}{2\sigma^2}\right) dx} = 2.8897\sigma \quad (12)$$

The average values of the noise parameters shown in Fig. 2c were obtained by dividing the antilog of the intercepts by 2.89.

To assess the detector-limited noise as a function of mass, we partitioned the mass axis into intervals containing a fixed number of mass channels. Each of the 24 resulting intervals contains approximately $10^5$ channels corresponding to a 50 kHz frequency band. This choice was made to match the apparent frequency intervals used internally by the Orbitrap processing software to estimate noise levels for the purpose of setting threshold levels (Details in Supplementary Notes 5 and 6). Histograms of the non-zero data elements in each of the intervals for the 50 pA dataset are overlaid in Fig. 2e. Only the low-intensity regime is shown to exclude larger ion-induced signals. The thresholds are plainly evident, and the shapes of the distributions are qualitatively the same as the shaded region of Fig. 2d. Crucially, the positions of the histograms progressively shift to a higher signal level as the mass increases. This reflects the influence of $1/f$ noise. The mean increment of Eq. (12) was estimated in each interval as the sample mean of the non-zero elements within the interval and the local average noise standard deviation in the $i^{th}$ interval $\sigma_i$ was obtained via division by 2.89. Given 24 mass-dependent values of $\sigma$, Eq. (11) can be used to estimate the overall noise parameters for all three datasets (Fig. 2f). The intercept and slope from linear regression are converted into the variance parameters $\sigma_W^2$ and $\sigma_F^2$, respectively, using the (squared) relationship of Eq. (12) (Details in Supplementary Note 1).

Given the mathematical expression for the frequency-dependent noise variance, it is a simple matter to compute its integral average over the full mass range. These values (Fig. 2f) agree within a few percent of the estimates based on the intercepts of the variance-vs-mean plots of Fig. 2c. Similar plots could be used to estimate the average noise in more chemically complex datasets if they include a sufficient number of low-signal peaks. Such estimates are likely good

enough to be used effectively in multivariate statistical analysis procedures.

The effect of operating the Orbitrap analyser in the non-linear regime is clearly seen in Fig. 2f, where the noise levels near intense spectral features become broadly and significantly inflated. Similar anomalies are evident in the 200 pA noise histograms in Fig. 2e. When trying to assess detection limits it is important to recognize that noise may be inflated near intense features (detailed description of unmodeled signal variations in Supplementary Note 3).

Table 1 shows all estimated noise parameters for the data acquired with different primary ion beam currents. The source-limited noise parameters $A$ and $R_N$ were obtained by fitting the lines shown in Fig. 1e, and the numbers in parentheses are statistical uncertainties from the fit. The detector noise parameters $\sigma_W$ and $\sigma_F$ were derived from fits to the lines in Fig. 2f, with data near high-intensity features excluded from the fit. Whilst statistical uncertainties are provided, they are small compared to uncertainties in the modelling assumptions needed convert censored means to variances, which are likely several percent. All parameters show an increase with beam current. It is notable, however, that the ratio of $A$ to $\bar{\sigma}$ is approximately 1.5, independent of beam current. As detailed in Supplementary Note 1, the detection limit depends solely on this ratio and the thresholding parameter. Based on the silver depth profiles, we find the detection limit to be 3.7 ions, where we define the detection limit to be the number of trapped ions required to achieve a 99.9% probability of observing a non-zero signal.

## Comparison between experiments and probabilistic model

The probabilistic model makes predictions about both the distribution of signal in a single mass peak as well as the second-order statistics of the mass spectrum as a whole. For a single mass peak with signal well in excess of the detector-noise level, the data are expected to be Gaussian with mean and variance given by Eq. (8). The situation is more complex when signal and noise have comparable magnitudes, and censoring is pervasive. The data are described by the WSoR distribution of Eq. (5) with zeroing of all data elements less than a threshold. Figure 3a compares the histogram of the 100 pA $^{107}Ag_4{}^{109}Ag_4Bi^+$ peak with the computed weighted-sum distribution function for a mean number of trapped ions equal to 3.61. The agreement is good, and both show that about 17% of the data elements have been censored. A decomposition of the probability function into its individual Rician components is shown in Fig. 3b.

The relationship between the variance and mean of a mass spectral peak has played a prominent role in our analysis, and the variance $vs$. mean curve can be predicted from the WSoR distribution. To examine the performance of our model over the entire range of signal intensities, statistical properties of the data were compared with a simulation based on the probabilistic model. All 192 of the measured peaks were simulated, and simulation "truth" was established by matching the predominant peak in each of the 34 clusters with the observed mean spectrum. Intensities of the remaining isotopic combinations were calculated using the natural abundances of Ag. A depth profile was obtained by making 1000 replicate simulations of the same base spectrum. Time-domain signals were constructed by summing abundance-weighted, zero-phase pure sinusoids and adding random noise. Signal sampling parameters, apodization and zero-padding were chosen to match the Orbitrap

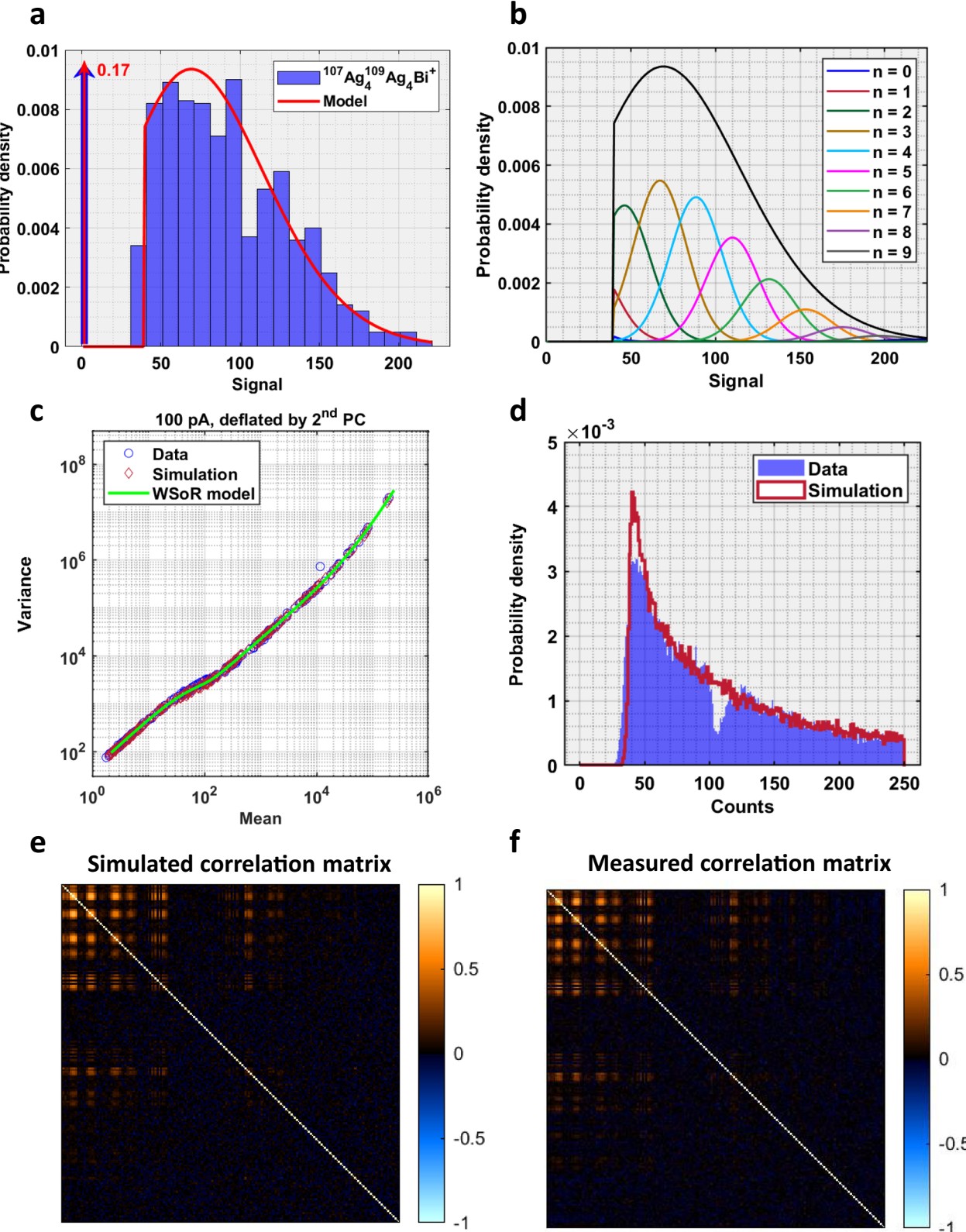

**Fig. 3 | Comparison of probabilistic model with experimental silver data.**
**a** Comparison of the histogram of the 100 pA $^{107}$Ag$_4$$^{109}$Ag$_4$Bi$^+$ peak with the computed weighted-sum distribution function for a mean number of trapped ions equal to 3.61. **b** Decomposition of the probability function into its individual Rician components. **c** Comparison of variance vs mean curves between observed and simulated 100 pA data. **d** Distributions of the data at low signal intensity. **e**–**f** Correlation matrices.

acquisition. Magnitude spectra were individually thresholded in 50 kHz blocks using estimates of the detector-noise standard deviation made in each block. The WSoR-predicted variance shows excellent agreement with the observed variance in Fig. 3c for the

100 pA dataset. One seeming outlier corresponds to Cs$^+$ and is discussed in detail in Supplementary Note 7. The WSoR model used the average value of the detector-noise standard deviation in this case as the low-intensity signals span the entire mass range. Additional

**Table 2 | Comparison of discussed methods for addressing heteroscedastic noise**

| Type | Method | Description | Comments |
|---|---|---|---|
| No scaling | - | - | Overweights higher intensity peaks relative to lower intensity ones by an amount proportional to their difference (unbounded). |
| Heuristic | Variance scaling | The noise variance vector is estimated as the sample variance. | The sample variance includes systematic spatial variation, which is erroneously treated as noise. It underestimates the significance of high-intensity over-dispersed peaks. |
| | Pareto scaling | The noise variance vector is estimated as the sample standard deviation. | The sample standard deviation includes systematic spatial variation, which is erroneously treated as noise. It underestimates the significance of low-intensity/censored peaks. |
| | Root-mean scaling | The noise variance is assumed to equal the global mean spectrum (i.e., the sample mean). | Overestimates the significance of low-intensity/censored peaks relative to high-intensity ones by an amount bounded by a constant factor. |
| | Log transform | Takes the logarithm of intensities in each spectrum. | From a noise equalisation standpoint, assumes noise variance is an exponential function of the signal. It is mathematically problematic for sparse datasets with a high fraction of zeros. |
| | Square-root transform | Takes the square-root of intensities in each spectrum. | Prone to overfitting the noise since each data element is assigned an individual estimate of its variance. These estimates can be exceedingly poor for low-intensity signals. |
| Machine learning-based | PFA | Directly estimates the uncorrelated noise variance vector as part of an iterative matrix factorization algorithm. | Erroneously characterises uncorrelated systematic variation as noise, can be very time consuming to compute, is susceptible to outliers. |
| Model-based (proposed in this work) | WSoR | The noise variance vector is computed based on a weighted-sum-of-Ricians statistical model. | |

Scaling methods divide each spectrum elementwise by the square root of the estimated or assumed noise variance vector.

comparisons between the WSoR model and the observed data are provided in Supplementary Note 1.

The distribution of the data at low signal intensity is compared in Fig. 3d. According to the data model, overdispersion is caused by all ions varying in tandem and results in the data being correlated. Simulated and data correlation matrices are compared in Fig. 3e, f. All diagonal elements in the images are unity. The simulation clearly provides a faithful representation of overall data distribution in this regime.

### Data scaling and transformations for multivariate analysis

Multivariate statistical methods[34] are useful for extracting the chemically relevant information pertaining to relatively few components embedded in a high-dimensional dataset. To better understand how the Orbitrap user community pre-process their data for multivariate analysis we studied the methods section of the 10 most cited papers (Web of Science) of the 244 papers that involve Orbitrap MS and "multivariate analysis" or "PCA" or "Principal Component Analysis" Of these, 8 papers gave no information on data scaling, 1 paper used Pareto scaling and 1 used natural Log transform. Very limited information was provided on the data pre-treatment prior to multivariate analysis. We also did a survey of the top 10 cited papers out of 35 found that also involved "metabolomic". One paper was common to both sets. Of these papers, 5 used Pareto scaling, 1 used Log transform, 1 used variance scaling and the remainder did not provide scaling information. Generally, more detailed accounts were provided of the data pre-treatment and reasons were given why scaling was necessary. However, no rationale was provided for using one scaling method over another. The list of papers considered is provided in Supplementary Note 7.

To evaluate different pre-processing methods, we will consider PCA[35,36] as the archetypal method for producing a low-dimensional representation of the data. Besides having well-known optimality properties, it is easy to compute. It should be kept in mind, however, that the PCA representation can be post-processed in various ways to obtain equivalent representations that might be more easily interpreted in physical terms. These include, for example, direct factor rotations[37] or employing a factored representation of the data in algorithms designed to produce non-negative components[38]. In addition, we will employ Probabilistic Factor Analysis (PFA). Like PCA, PFA produces a low-dimensional factor model of the data but has the added advantage that it estimates the uncorrelated part of the statistical noise[39–41].

All preprocessing methods discussed in this paper are summarized in Table 2. In general, scaling methods seek to make the noise uniform by multiplying the data matrix by the inverse matrix square root of the noise covariance matrix. The methods described differ only in the manner in which the noise covariance is approximated.

Before considering the full probabilistic model derived in this paper, it is instructive to examine the simpler model with error covariance given by Eq. (8), which is suitable for moderate to high signal levels. Here, the noise covariance matrix is a symmetric rank-1 modification of a diagonal matrix. Weighting involves the inverse of the matrix, which can be computed[42], and after some algebra:

$$\mathbf{C}_E^{-1} = \text{diag}(A E[\mathbf{s}])^{-1} - \frac{1}{A^2}\left(\frac{1}{R_N^2} + \frac{\mathbf{1}^T E[\mathbf{s}]}{A}\right)^{-1}\mathbf{1}\mathbf{1}^T \quad (13)$$

This matrix is a rank−1 modification of the diagonal matrix whose (uncorrelated) elements are the inverse of the scaled mean spectrum. In the limit that $R_N$, the relative standard deviation due to overdispersion goes to zero (i.e., the data are Poisson-like), only this diagonal term survives. Considering the off-diagonal elements, the right-hand term within the parentheses is the total number of ions in the trap and for moderate overdispersion, both terms are large numbers of which we take the inverse sum. For the cases considered here, the off-diagonal elements are effectively zero and estimates of the uncorrelated variance are sufficient for scaling purposes.

### Application to an OrbiSIMS image of a drosophila central nervous system

To compare the effectiveness of the different scaling methods in the context of a biological tissue, we used the well-characterised larval central nervous system (CNS) of the fruit fly, *Drosophila melanogaster*[43]. An OrbiSIMS imaging dataset was acquired from transverse cross sections of the thoracic region of the larval CNS (experimental and analytical details are in Methods and

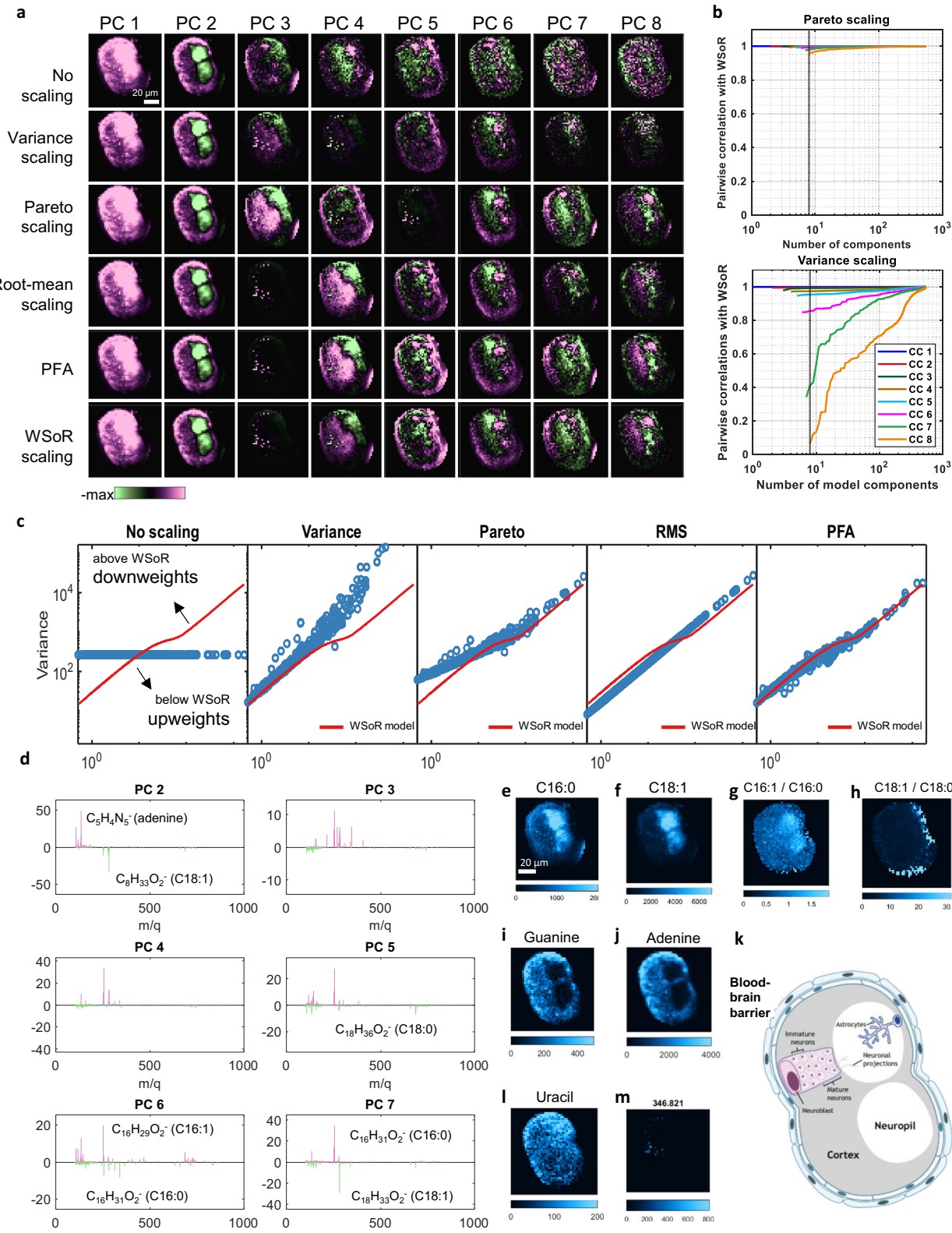

**Fig. 4 | Effect of scaling bias on multivariate analysis of an OrbiSIMS drosophila CNS image. a** PCA scores (PCs 1 to 8) of the *Drosophila* CNS dataset using various scaling methods purple = positive, green=negative, the absolute maximum value in the scale is clipped for better contrast, the scales vary from +/− of this for each PC. **b** Correlations between the Pareto-scaled model (top) and Variance-scaled model (bottom) with the 8-component WSoR model. **c** Assumed variance vs. mean plots for different scaling methods compared to WSoR model. **d** WSoR loadings of PCs 2-7. Intensity distributions of **e** C16:0, **f** C18:1, **g** ratio C16:1/C16:0 **h** ratio C16:1/C16:0, **i** adenine [$C_5H_4N_5$]⁻ and **j** guanine [$C_5H_2N_5$]⁻. **k** Schematic showing cross section of the *Drosophila* larval CNS with specific cell types indicated. Intensity distributions of **l** uracil [$C_4H_3N_2O_2$]⁻ and **m** *m/z* 346.821.

Supplementary Note 8). The dataset consists of 553 mass peaks spanning a $m/q$ range between 100 u and 863 u within an ROI of 1165 pixels. The eigenvalue plot showed a sharp break suggesting 8 chemical components are significant with respect to noise (Supplementary Fig. 18). In agreement with our observation for the silver data, no scaling is not effective in separating signal from noise in the CNS dataset (Fig. 4a). PFA and WSoR clearly find the same 8 distinct spatial patterns. Since all of the features contain high-intensity peaks, we would expect Pareto scaling to work well and, in fact, this method discovers the same 8 components as WSoR, although they are reordered. Root-mean scaling also finds these 8 components, although PCs 7 and 8 are noticeably noisier. This observation is consistent with the idea that root-mean scaling relatively upweights low-intensity peaks, which, in this case, represent noise. No scaling and Variance scaling do not perform as well, each capturing only 6 of the WSoR components within an 8-component model. Interestingly, the 6 components are not the same. Variance scaling, which down-weights high-intensity peaks, still finds the relatively low-intensity peaks associated with the speckle pattern of WSoR PC 3. This contrasts with no scaling, that down-weights the low intensity peaks and does not find the speckle pattern within the first 8 PCs. The similarity of two low-rank models can be assessed by computing the principal angles and principal vectors (principal scores here), or canonical correlations, between the two subspaces[44]. Correlation greater than about 0.7 is required for two score images to unambiguously describe the same spatial features. The top panel of Fig. 4b shows the correlations between the Pareto-scaled model and the 8-component WSoR model as the dimension of the Pareto-scaled model is varied. The correlations are all close to 1, indicating both models are describing the same 8 components. This contrasts with the bottom panel, which shows the evolution of Variance-scaled model. As noted earlier, the 8-component Variance-scaled model agrees with the WSoR model in 6 of the components. The bottom panel of Fig. 4b suggests that several tens of components will be required before the Variance-scaled model captures the information in WSoR PCs 7 and 8 in a recognizable form. This has been validated by plotting the principal scores as a function of model size (Supplementary Fig. 19). Figure 4c shows how the estimated variances in the different scaling methods down-weight or upweight the signals at different intensities.

PCA with either WSoR, PFA, Pareto or root-mean scaling helps provide biological interpretation. PC 2 (Fig. 4a, d) separates the cortex (which shows a strong signal for nucleobases) from the neuropil (which shows a strong signal for lipids). WSoR PC 7 (Fig. 4a, d) reveals that C16:0 and C18:1 are anticorrelated with different spatial distributions. We find that the signal intensity ratio of saturated to unsaturated fatty acids C16:1 to C16:0 is enhanced in the neuropil whereas the ratio C18:1 to C18:0 (Fig. 4e–h) is stronger at the periphery of the cortex along the blood-brain barrier. In addition, nucleobases are detected as a result of DNA/RNA fragmentation. the strongest signals for adenine ($C_5H_4N_5^-$) and guanine ($C_5H_3N_5O^-$) map close to the surface of the cortex. This corresponds to the immature neuron layer where neural cell bodies are smaller and packed more tightly than in the inner mature neuron layer (Fig. 4i, j). The signal for uracil ($C_4H_3N_2O_2^-$), however, is not strongly enriched in the cortex relative to the neuropil (Fig. 4k) as this nucleobase is incorporated into RNA not DNA, which does not accumulate in the nucleus and so is more uniformly distributed across the whole CNS (Fig. 4l). PC 6 separates some lipids marking the neuropil and a superficial layer of cells surrounding the CNS localised to the glia of the blood-brain barrier (Fig. 4m).

The PC 3 scores image (PC5 using Pareto) contains speckles that do not obviously correspond to anatomical or cellular landmarks in the CNS. These correspond to signals from the interface between the CNS section and the indium tin oxide on slide substrate (e.g., $m/q$ 346.8208, Fig. 4m). These likely result from differences in sample thickness or pinholes across the section. Full PCA results can be produced with the

code published, and a table with relevant peaks is in Supplementary Note 8.

## Application to an OrbiSIMS image of a mouse testis section
Experimental details are in **Methods** and Supplementary Note 8. The image contains 9900 pixels and records intensities in 163 mass-peak intervals. This data is an example of low intensities where censoring to remove detector noise dominates. Consequently, the data in this regime appear Poisson-like as previously discussed.

Figure 5a compares PCA scores images and the estimated variance (Fig. 5b) against the WSoR model for each scaling method as before. The spatial features observed in the scores images (Fig. 5a) correspond closely to linear combinations of mass peaks found in the raw data (Fig. 5c-l). Analysis of the eigenvalues (Supplementary Fig. 18) indicates there are 7 information components. The spatial pattern in PC2 is interpreted as lipid droplets, which are normally found in the interstitial spaces between seminiferous tubules. The $C_{41}H_{79}O_{12}S^-$ pattern described in PC2 (Fig. 5c, e, f) corresponds to seminolipid (sulfogalactosylglycerolipid) that plays an important role in male reproduction[45].

For this data, Pareto works poorly since it down-weights low intensity ions. WSoR PC 4 pattern corresponds to $C_4H_8N_3O_2^-$ and $C_4H_6N_3O^-$ (Fig. 5i, j) and the WSoR PC 7 pattern corresponds to $KCl_2^-$ (Fig. 5l exposed pre-treated substrate). With Pareto scaling these patterns are not found (Fig. 5a) until PC 34 and PC 77 and with no scaling at PC 39 and PC 81. For Poisson-like data, root-mean scaling is appropriate so that the performance is similar to WSoR (and PFA). This illustrates that the correct scaling produces parsimonious models that order the components by chemical information content, leading to improved interpretability. WSoR PC 6 also identifies an Orbitrap instrumental artefact where some peaks with weak intensity (e.g., isotopologues) fall below the threshold in some pixels and are censored, causing a non-linear effect. This is important information for studies using isotopic labelling strategies and highlights the effectiveness of WSoR scaling method to separate noise from other sources of information.

## Application to an Orbitrap-DESI image of a rat liver section
To demonstrate that the WSoR model is applicable to other ion sources, we have analysed a desorption electrospray ionisation (DESI) image collected using a Q-Exactive Orbitrap mass analyser (Fig. 6a–c). The ion source is effectively an electrospray type with the sample, in ambient conditions, located between the source and the mass spectrometer inlet. The data set is previously published and the details of rat liver section, embedding and the Orbitrap-DESI experimental details are described elsewhere[46]. A subset of 30 peaks are selected to test the scaling methods including spatial features that in group B are clustered in specific locations, with mean intensities in the region of low intensity dominated by detector noise and signal censoring (16 peaks) and in group A are distributed across the tissue and with mean intensity in the high intensity region dominated by counting statistics (14 peaks).

Analysis of the eigenvalues (Supplementary Fig. 18) indicate that there are 7 chemical components. PFA of group A and group B peaks separately (Fig. 6b) finds 6 chemical components plus an additional component that related to thresholding of a carbon isotope: at low intensities, the two isotopes experience different levels of thresholding, which induces an additional component.

With no-scaling and Pareto scaling, which both down-weight low intensities, we find WSoR component 5, which maps to Group B, in components 10 and 16, respectively. Variance scaling, which down-weights high intensities, finds a Group B component in PC 21. Root-mean scaling is complicated, finding Group B and Group A components in PCs 15 and 10, respectively.

This dataset provides a practical example featuring significantly low-intensity features (like in the testis dataset, Fig. 5) and significantly high-intensity features (like the CNS dataset). Biases introduced by no-scaling, variance, Pareto, and root-mean scaling are clearly observed

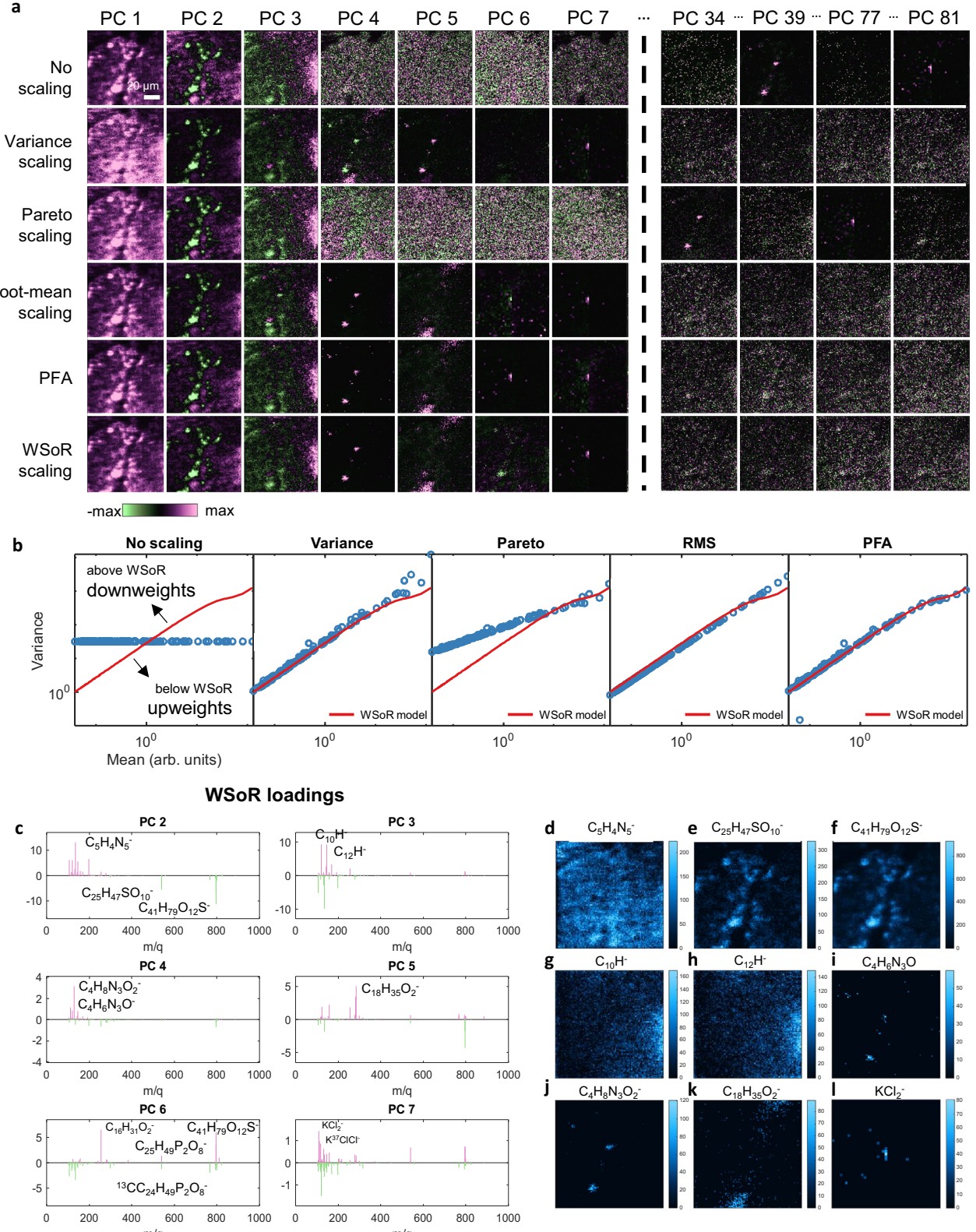

**Fig. 5 | Effect of scaling bias on multivariate analysis of an OrbiSIMS mouse testis image. a** PCA scores (PCs 1-7, 34, 39, 77, 81) of the mouse testis dataset using various scaling methods (purple=positive, green=negative, the absolute maximum in the scale is clipped for better contrast, the scales vary from +/− of this for each PC). **b** Assumed variance vs. mean plots for different scaling methods compared to WSoR model. **c** WSoR loadings of PCs 2–7. Intensity distributions of **d** $C_5H_4N_5^-$, **e** $C_{25}H_{47}SO_{10}^-$, **f** $C_{41}H_{79}O_{12}S^-$, **g** $C_{10}H^-$, **h** $C_{12}H^-$, **i** $C_4H_6N_3O^-$, **j** $C_4H_8N_3O_2^-$, **k** $C_{18}H_{35}O_2^-$ and **l** $KCl_2^-$. The WsoR PC 7 scores map (and equivalent for other scaling and ion distributions) has been cropped to a smaller number of pixels to facilitate visualisation.

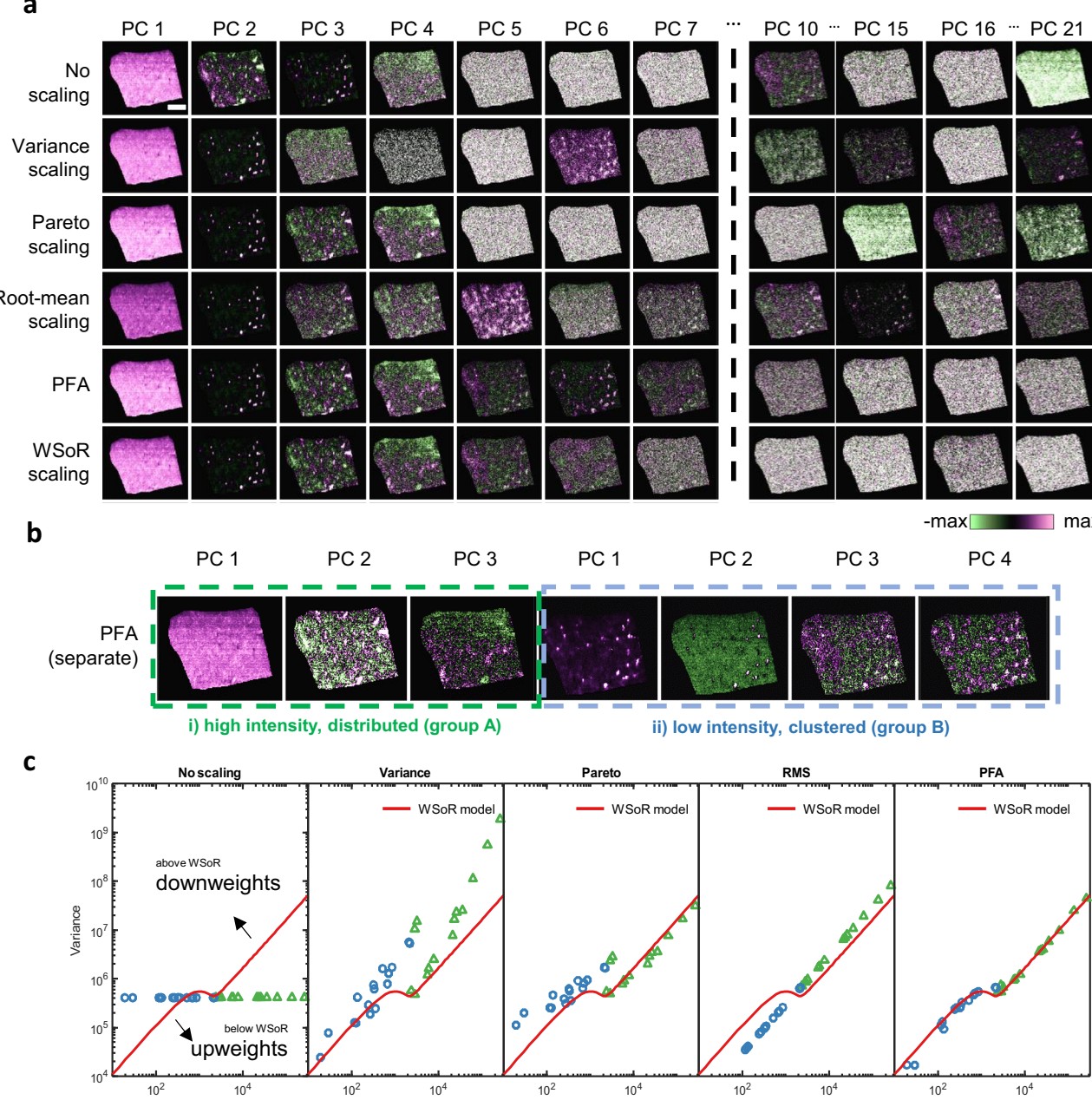

**Fig. 6 | Effect of scaling bias on multivariate analysis of an Orbitrap-DESI rat liver section image. a** PCA scores (PCs 1-7, 10, 15, 16, 21) of the rat liver dataset using various scaling methods (purple=positive, green=negative, the absolute maximum in the scale is clipped for better contrast, the scales vary from +/− of this maximum in the scale is clipped for better contrast, the scales vary from +/− of this for each PC). **b** PFA done in two separate sets of ion images with high intensity, distributed across all pixels (group A) and low intensity, in clusters of pixels (group B). **c** Assumed variance vs. mean plots for different scaling methods compared to WSoR model.

with chemical components relegated to PCs beyond the expected number of chemical components. WSoR in contrast provides a parsimonious analysis.

## Discussion

We have developed a comprehensive model based on a weighted sum of Rician distributions (WSoR) that describes data generation and noise that accounts for variations in secondary ion production, efficiency of ion transfer, mass spectrometry with a high-field Orbitrap, and data processing. Model construction involved a nested series of conditional probability problems that describe how variation is compounded at each step of the measurement process.

At relatively high signal levels where detector noise is negligible, ion-counting statistics govern the variatio,n and the signal can be parameterized in terms of an ions-to-signal scale factor and a variance term that accounts for overdispersion, if any, with respect to Poisson-like behaviour. Variation in mass channels that do not correspond to oscillating ions is caused by detector noise. This can be described in terms of a frequency-independent white-noise component plus fluctuation noise that increases with mass. The low-signal statistics are dominated by censoring, where only signals exceeding a threshold are recorded. Procedures to estimate the model parameters from a simple depth profile were detailed, and a simulation implementing the model was constructed. The simulation reproduced all general

features of the data in a self-consistent manner, including the second-order statistics.

A major impetus behind the development of the noise model was to provide a principled basis for reducing noise bias in multivariate statistical analysis. In such analyses, we assume the sought-after chemical information resides in a low-dimensional subspace of the full dataset, and the goal is to separate the chemically relevant components from those representing only noise. We evaluated the effectiveness of our WSoR method for three biological imaging data sets and compared the performance with no-scaling, variance, Pareto and root mean square scaling. The Drosophila CNS and mouse testis are OrbiSIMS images, whilst the third is an independent published[46] image data set using a DESI source. These provide examples where intensities are low and are affected by detector noise and censoring, intermediate, and include statistics from the ion generation process to high intensities. The DESI example demonstrates that the WSoR is generalizable to other ion sources and validates the model for independent Orbitrap measurements.

We used the machine learning method, Probabilistic Factor Analysis, to estimate the noise directly from experimental data to validate the WSoR model. PFA estimates spatial and spectral factors and the uncorrelated (i.e., Poisson-like) part noise over the entire range of intensities. In each example, we show that WSoR and PFA have good agreement.

We provide a framework to understand the variable performance of the different scaling methods. In the Drosophila CNS example, variance scaling down-weights intense ions and performs poorly whereas Pareto scaling performs similarly to WSoR. In contrast, in the testis example, Pareto scaling down-weights low intensity ions and performs poorly whilst variance scaling more closely matches WSoR. In the DESI data set, variance scaling and root-means scaling down weight the intense peaks and Pareto scaling downweights less intense peaks so that spatial features move well beyond the 7 chemical components. Root-mean scaling has a complicated representation.

In all of these examples, we find that the WSoR provides noise-unbiased scaling allowing the chemical information to be captured efficiently in the leading Principal Components and separated from the noise. In contrast, the performance of variance scaling, Pareto scaling and RMS is variable, being good and poor on a case-by-case basis, depending on the distribution of ion intensities. As expected, no-scaling is always poor. Log-transform and square-root transform were also analysed but is not considered further since it is always very poor and should not be used (details in Supplementary Note 8). In fact, it is unclear how to implement a log transform for highly sparse data with many zero data elements in a way that gives proper weight to the zeros given that zeros in mass spectrometry are meaningful – they indicate the absence of a particular ion.

These examples are for imaging datasets but since there is a full mass spectrum per pixel, the method is equally applicable to data structured in any form or individual sets of spectra, for example, in liquid chromatography mass spectrometry. In this study, we measure the noise threshold parameter for three high-field Orbitrap instruments and find the noise threshold parameter is instrument-specific (Fig. 7). We also determine the scaling parameter, $A$, which allows the arbitrary unit Orbitrap intensity scale to be calibrated to a true count scale. This allows the sensitivity to be compared with other mass spectrometers[47]. The WSoR models for the two OrbiSIMS instruments are different because one of them was purposefully operated with a lower noise thresholding parameter. Analysts can use our method to generate their own WSoR to give estimates of the noise covariance for scaling in multivariate analysis

## Methods

This research complies with all relevant ethical regulations. Animal studies were performed under a UK Home office approved project

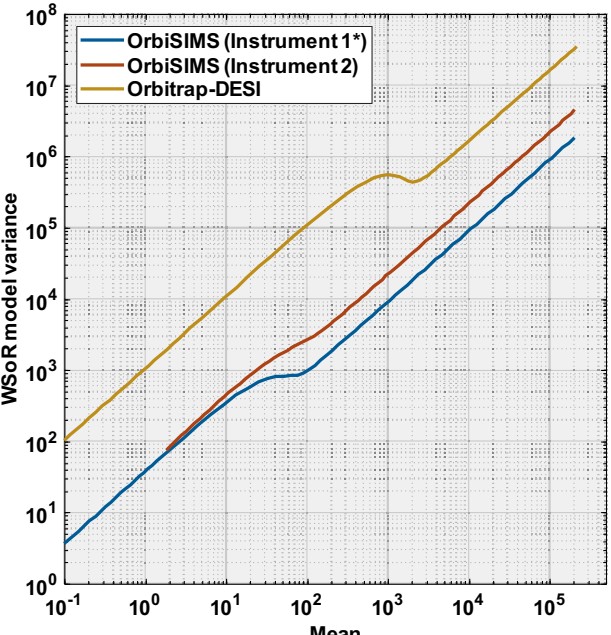

**Fig. 7 | Comparison of the WSoR models from the three instruments used in this study.** *The OrbiSIMS Instrument 1 was configured to have a lower censoring threshold than the factory standard.

license (PAA689E24) and in accordance with institutional welfare guidelines and local ethical committees. All efforts were made to ameliorate any suffering. Extended methods are described in the Supplementary Information file.

### OrbiSIMS of silver

Datasets were acquired using an OrbiSIMS instrument (Hybrid SIMS, IONTOF GmbH). Depth profiles were obtained in a single beam mode using a 30 keV $Bi_n^+$ ion beam from a liquid metal ion source. The average primary ion beam currents were 50 pA, 100 pA and 200 pA. The beam was set to scan an area of 320 μm × 320 μm of which secondary ions were collected from the central 200 μm × 200 μm of the crater. 70 × 70 pixels were used per scan, which resulted in a single injection per scan. The analyser extraction voltage was set at 2000 V The Orbitrap mass analyser was operated with 512 ms injection time and 512 ms transient time for a mass resolving power of 240,000 at m/q 200. A total of 1000 scans were acquired for each beam current. For all OrbiSIMS analyses, the sample target potential was set to 57.5 V and the He collision cell pressure was kept at 0.12 bar in high collisional cooling mode.

### OrbiSIMS of Drosophila CNS

*Drosophila* larvae ($w^{1118}$ *iso31* strain) were raised on a yeast-cornmeal diet (66.3 g/L cornmeal, 58.5 g/L glucose, 23.4 g/L dried inactivated yeast, 7.02 g/L agar, 1.95 g/L Nipagin, and 7.8 mg/L Bavistan) until late L3 stage. A larval central nervous system (CNS) was dissected in phosphate buffered saline (PBS) and fixed in 4% paraformaldehyde in PBS for 30 min at room temperature. After fixation, the sample was washed in PBS and stained with 13.1 μM Toluidine Blue O for 30 min at room temperature, followed by a PBS wash. The CNS was then embedded in 4% carboxymethyl cellulose (sodium salt) in PBS and flash frozen in a bath of dry ice and 2-methylbutane. Larval CNS transverse sections were cut using a Leica CM3050 S Cryostat (Leica Microsystems, Wetzlar, Germany) set to 10 μm thickness, and −20 °C chamber and object temperatures. Tissue slices were thaw mounted onto an ITO-coated glass slide (25 mm × 25 mm) with a resistivity of 70–100 Ω/sq (Sigma-Aldrich, #703176), cleaned with sequential washes in 70:30 acetone:water, 2:1 chloroform:methanol and then

hexane, and subsequently vacuum packed and stored at −80 °C until analysis. Before unsealing the sample, it was allowed to warm to room temperature for 20 mins to reduce condensation on the sample surface. The CNS section was then analysed on an OrbiSIMS (Hybrid SIMS, IONTOF GmbH) instrument. The analysis was performed at ~25 °C using a 20 keV $Ar_{3500}^+$ quasi-continuous GCIB analysis beam with a spot size of ~3 μm using a sawtooth raster mode, a current of ~12 pA, a duty cycle of 15 % and a cycle length of 200 μs. The total ion dose the image was $8.88 \times 10^{14}$ ions cm$^{-2}$ with 14,800 shot per pixel. The image area was 250 μm × 250 μm, with a 5 μm pixel size. The Orbitrap MS was operated in negative-ion polarity with a mass resolution of 240,000 @ 200 $m/q$ and an injection time of 2961 ms, with the automatic gain control switched off. The surface potential was 6.1 V. The collisional cooling pressure was set to low with a pressure of $4.3 \times 10^{-2}$ mbar. Mass spectral information was acquired for the range $m/q$ 100–1000.

## OrbiSIMS of mouse testis

Male mice (strain FVBN/J, wildtype, 6–8 weeks) were housed in a temperature-controlled room at 21 °C with a 12-hour light: dark cycle. Water and food were provided *ad libitum*. Mice were killed by cervical dislocation and testes dissected and frozen in liquid nitrogen and stored at −80 °C before sectioning. Mouse testis sections were attached to the cryotome chuck using water and blue roll tissue and cut using a Thermo Scientific NX70 Cryostar (Thermo Scientific, Runcorn, United Kingdom) set to 10 μm thickness, with ~-25 °C knife and ~-10 °C object temperatures. Tissue slices were thaw mounted onto an ITO-coated glass slide (25 mm×75 mm) with a resistivity of 70–100 Ω/sq (Sigma-Aldrich, #576352), and subsequently vacuum packed and stored at −80 °C until analysis. Before unsealing the sample, it was allowed to warm to room temperature for 10 mins to reduce condensation on the sample surface. The testis section was then analysed on an OrbiSIMS (Hybrid SIMS, IONTOF GmbH) instrument. The analysis was performed at ~25 °C using a 20 keV $Ar_{2500}^+$ quasi-continuous GCIB analysis beam with a spot size of ~5 μm using a sawtooth raster mode, a current of ~1.44 pA, a duty cycle of 10 % (lower to previous experiments due to a misalignment) and a cycle length of 200 μs. The total ion dose density of the image was $1.1 \times 10^{14}$ ions cm$^{-2}$ with 2500 shot per pixel. The image area was 200.00 μm × 200.00 μm, with a 2 μm pixel size. The Orbitrap was operated in negative-ion polarity with a mass resolution of 240 000 @ 200 $m/q$ and an injection time of 511 ms, with the automatic gain control switched off. The surface potential was -30 V. The collisional cooling was set to low with a He pressure of $4.2 \times 10^{-2}$ mbar. Mass spectral information was acquired for the range $m/q$ 100 –1500 u.

## Reporting summary

Further information on research design is available in the Nature Portfolio Reporting Summary linked to this article.

## Data availability

All OrbiSIMS data reported in this manuscript have been deposited as imzML or MATLAB.mat files to figshare (https://doi.org/10.6084/m9.figshare.28684622). imzML files can be processed in MATLAB using the imzML parser of SpectralAnalysis (https://github.com/AlanRace/SpectralAnalysis). Source data are provided with this paper.

## Code availability

The codes for performing PCA of biological datasets alongside the data matrices for Figs. 4 to 6 have been deposited in Code Ocean (https://doi.org/10.24433/CO.7275815.v1).

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

## Acknowledgements

The authors are grateful to Professor Maurice Cox at the National Physical Laboratory and Dr Stewart A Jones at Astra Zeneca for helpful discussions. This work was part-funded by The UK National Measurement System. NPL is operated by NPL Management Ltd, a wholly-owned company of the UK Department of Science, Innovation and Technology. CLN, GFT, ISG and APG were supported by a Wellcome Technology Development Grant (104566) and APG by a Wellcome Investigator Award (223760). YJ, CLN and APG were also supported by the Francis Crick Institute, which receives its core funding from Cancer Research UK (CC2101), the UK Medical Research Council (CC2101) and the Wellcome Trust (CC2101). JB was supported by the CRUK Rosetta Grand Challenge grant (A25045 to JB) and we thank Mariia Yuneva at the Francis Crick Institute for the provision of the mouse testes sample. For the purpose of Open Access, the authors have applied a CC BY public copyright license to any Author Accepted Manuscript version arising from this submission.

## Author contributions

All authors contributed to the manuscript and approved the final manuscript. The concept was devised by M.R.K., G.F.T. and I.S.G. M.R.K. developed the statistical methods with G.F.T. and input from I.S.G., A.M. and K.A., A.M. and K.A. provided insight into the instrument and algorithmic processes in Orbitrap mass spectrometry. A.P. and H.A. provided insight into details of the OrbiSIMS instrument operation. A.P., G.F.T., R.H., J.Z. and L.M. acquired OrbiSIMS silver data. C.L.N. and Y.J. acquired OrbiSIMS images of Drosophila samples provided by A.P.G.. C.L.N., Y.J. and A.P.G. interpreted the Drosophila data. A.E. acquired imaging data of the mouse testis sample provided by J.B. and the CRUK Rossetta grand challenge consortia. A.E., C.L.N., Y.J. and J.B. chemically interpreted the mouse testis data. A.D., R.G. and Z.T. measured and provided an interpretation for the DESI image of a rat liver section. I.S.G. devised the original experiments and oversaw the study.

## Competing interests

The authors declare the following competing financial interest(s): AM, KA are employees of Thermo Fisher Scientific, the manufacturer of Orbitrap instrumentation used in this research and AP, HA are employees of IONTOF, the manufacturer of OrbiSIMS mass spectrometer used in this research. The remaining authors declare no competing interests.
