## [Transparent Peer Review file · Nature Communications]

Orbitrap noise structure and method for noise unbiased multivariate analysis

Corresponding Author: Professor Ian Gilmore

Version 0:

Reviewer comments:

Reviewer #1

(Remarks to the Author)

This publication presents a new scaling method for orbitrap data generated using a secondary ion mass spectrometer (SIMS). The untargeted nature of SIMS analysis provides a wealth of data that is not straightforward to interpret, especially on complex organic samples. The combination with a high resolution mass analyzer such as an orbitrap aids in identifying the mass signals but even increases the data load, further complicating manual data analysis in order to identify significant changes in the data. Multivariate analysis has become an invaluable tool to do just that but due to the way the MS analysis is performed, data scaling is necessary to gain information on higher mass, biologically relevant molecules. Several scaling methods are available but it is not at all obvious which one is most suitable for any given dataset, especially novel ones such as generated by an OrbiSIMS. Therefore, studies like this are very important to facilitate biological discoveries hidden in the dataset.

Major comments:

This publication is heavy on the introduction and establishing the mathematical foundation for their newly developed scaling method. At the beginning in the abstract the authors mention life sciences which lets one assume that this will be the focus of the publication and/or the intended application of the algorithm. At the end they do come around to show "life-science"-results but while it seems to work excellently to simulate inorganic data, they fail to demonstrating major benefits of, or improvements on existing scaling methods for the biological samples shown. So, while this work is by no means uninteresting, the results are currently lackluster. My recommendation is to focus solely on the theoretical background and the inorganic data in this publication, while further exploring biological applications and publish those at a later date, provided that significant improvements are found. Alternatively, provide an outlook for further developments to improve biological data analysis.

Minor comments:

Page 16, line 429: Please list the papers, also I assume the most cited publications are older publications, maybe more recent studies contain more information on data scaling.

Page 20, line 508: Rephrase, do you mean: all scaling and log transformation are effective...

Line 513: "Pareto scaling... ..may be less robust for complex, biological OrbiSIMS imaging datasets." Is this assumption supported by your experimental?

Figure 5j: The figure is too small to be useful, add labels for the highest loadings or add a bigger version to the supplementary information, the loadings in the SI are too small as well.

Line 518 ff: The references to the Figure 5 b-h are mixed up in the entire paragraph.

Page 20 line 540: "ether-linked Pes" change to PEs

(Remarks on code availability)

Reviewer #2

(Remarks to the Author)

Keenan et al. propose a theoretical model of the noise generated by the Orbitrap mass analyzers in the low intensity region.

For modeling, they use data generated by SIMS which provides a consistent input ion current with known noise properties. Moreover, they propose a new method (WSoR) for data rescaling that capitalizes on the knowledge of the noise model.

The manuscript strengths are in the theoretical work to mathematically model the noise. The manuscript weaknesses are in the validation and in demonstrating the practical impact of the proposed theory.

1. A key concern is about the validation of the proposed WSoR method. The authors do it in three ways: through data simulation (Figure 4), through discovery of the Cs+ contamination in first N principal components, and through application to an imaging dataset. However, the simulation is still a theoretical example and the practical use and impact for discovering the Cs+ contamination is not clear. Re application to an imaging dataset, Figure 5i shows no improvement for WSoR compared to “no scaling” in terms of discovering spatial patterns. For each spatial pattern (what the authors call “CNS pattern”) shown in PCA scores for WSoR-transformed data, there is a similar spatial pattern visible in the PCA scores in “no scaling” results.

2. It's not quite clear what's the basis of the molecular interpretations in Figure 5 because of two factors. First, there are no details on how the molecular identification was done. The authors should briefly comment on the methodology used for the reported molecular assignments. If they used the accurate m/z matching, they should share the details on the database (if any), the m/z tolerance used, and the m/z delta between the theoretical m/z and the observed one. It may additionally be worth mentioning that annotations of plasmalogen lipids and nucleotides are commonly the result of fragmentation reactions rather than intact detection. Moreover, the reported PE-Cer(34:2) is isobaric to PA(34:1) which can be a fragment of larger lipids. Second, the claimed correspondence between the mass spectrometry signals and cell types is not supported with any evidence.

3. A brief inspection of the raw data behind Figure 5 raises some questions about the interpretation (lines 501-553), especially the fatty acid analysis (subfigure 5b). Looking at individual images and intensity ratios, the statement about polyunsaturated vs saturated fatty acids does not appear to hold up. The pattern shown in 5b is caused mainly by the fact that palmitic acid (16:0) has a very different distribution from the other peaks. The inclusion of C14 fatty acids is somewhat redundant as only 14:0 and 14:1 are present, and a full order of magnitude below the corresponding C16 and C18 signals. Similarly, 16:2 does not contribute meaningfully to the sum, meaning that the image is indistinguishable from 18:2 / (18:0+16:0). In this set, I would argue that 18:2 and 18:0 are strongly correlated, which is the opposite observation from what the authors discuss! This also is readily apparent from a quick manual inspection, and does not really provide a good example of multivariate analysis used for hypothesis generation.

4. Within the imaging dataset analysis (lines 501-553), figure panels are referred to with the wrong figure number, e.g. references to 5b as 5d on line 519; 5e as 5g on line 530; 5g as 5h on line 539. The data in 5h (ratio of C18 to C16 fatty acids) is not referred to anywhere.

5. The testis analysis in supplementary note 9 is not mentioned in the main text, but is listed as the main individual contribution of two of the co-authors. The authors may want to consider including a short statement about that analysis and the value it adds to the manuscript.

(Remarks on code availability)

There are no installation instructions provided and no README file

Reviewer #3

(Remarks to the Author)

(Remarks on code availability)

Version 1:

Reviewer comments:

Reviewer #1

(Remarks to the Author)

I appreciate the extensive responses to my comments and the additional experiments. My concerns have been addressed and the significance of the model for biological data-sets is way clearer now. In my opinion, no additional experiments are needed.

I only have a few minor comments:

Figure 5a WSoR PC6 seems to highlight a feature captured by no other scaling method. Is this anything of interest? Line 572: why not test on the full data-set? Is it too computationally demanding? The 2 other bio data-sets use 100-500 peaks.

Line 573ff: What do mean with group C/D, this is not explained anywhere in the text or SI. Is there a group A/B? This section needs to be slightly reworked.

(Remarks on code availability)

Reviewer #3

(Remarks to the Author)

(Remarks on code availability)

Orbitrap noise structure and method for noise-unbiased multivariate analysis

We are very grateful for the reviewers' comments and for the opportunity to submit a major revision. We are sorry for the delay but it took time to do a substantial revision including an additional biological data set using desorption electrospray ionisation (DESI) as the ion source.

We have taken into account the reviewer's request to focus more on biological data to demonstrate the effects of the different scaling methods. We have now removed Figure 4 using silver data and focus on three biological data sets (1) drosophila CNS, (2) mouse testis – significantly expanding our previous analysis and adding a new main figure and (3) DESI image of rat liver detailed in (A. Dannhorn et al, *Anal. Chem.* 2020, 92, 16, 11080–11088) including a new main figure. The three data sets cover examples of high intensities, low intensities and high to low intensities. We have developed a much clearer way to show how no-scaling and the commonly used scaling methods variance, Pareto, root-mean scaling (RMS) upweight and down-weight ions, which introduces bias into principal component analysis (PCA). The objective of PCA is to parsimoniously capture the chemical components with separation from noise components. We clearly show that our Weighted Sum of Ricians (WSoR) based on our theoretical framework always provides the most efficient solution, whereas the other scaling methods perform variably being sometimes good or poor depending on the statistical detail of the data.

Furthermore, the addition of the DESI data set demonstrates the generality of our WSoR scaling method to a completely different ionisation source and is also data from an independent laboratory. From our understanding of the literature and discussions with the community there is, at present, no principled way to select a scaling method and so it is a matter of luck if they work well or not. With the growing use of computational methods on ever larger data sets, we therefore think that our WSoR method is important for improving the reliability and effectiveness of Orbitrap data analysis.

Recently, we have been conducting an international interlaboratory study to look at noise in OrbiSIMS instruments and determined their WSoR functions. We received data for 8 instruments including a newer Orbitrap Exploris design and all exhibited excellent agreement with our WSoR model. This will be reported separately later but we are happy to share a plot in confidence, if requested.

In the following we provide a point-by-point response to the reviewers.

Reviewer #1

This publication presents a new scaling method for Orbitrap data generated using a secondary ion mass spectrometer (SIMS). The untargeted nature of SIMS analysis provides a wealth of data that is not straightforward to interpret, especially on complex organic samples. The combination with a high resolution mass analyzer such as an orbitrap aids in identifying the mass signals but even increases the data load, further complicating manual data analysis in order to identify significant changes in the data. Multivariate analysis has become an invaluable tool to do just that but due to the way the MS analysis is performed, data scaling is necessary to gain information on higher mass, biologically relevant molecules. Several scaling methods are available but it is not at all obvious which one is most suitable for any given dataset, especially novel ones such as generated by an OrbiSIMS. Therefore, studies like this are very important to facilitate biological discoveries hidden in the dataset.

We are very grateful for the reviewer's supportive comments on the value of our work, especially selecting the most suitable scaling method for biological data. We agree that it is not obvious which scaling method to use and we show that our WSoR method works best in all cases.

Major comments:

This publication is heavy on the introduction and establishing the mathematical foundation for their newly developed scaling method. At the beginning in the abstract the authors mention life sciences which lets one assume that this will be the focus of the publication and/or the intended application of the algorithm. At the end they do come around to show "life-science"-results but while it seems to work excellently to simulate inorganic data, they fail to demonstrating major benefits of, or improvements on existing scaling methods for the biological samples shown. So, while this work is by no means uninteresting, the results are currently lackluster. My recommendation is to focus solely on the theoretical background and the inorganic data in this publication, while further exploring biological applications and publish those at a later date, provided that significant improvements are found. Alternatively, provide an outlook for further developments to improve biological data analysis.

The statistics of the Orbitrap measurement process are complex and it has been a long and difficult task to work it all out. So, we are therefore very grateful for the reviewer's appreciation of the mathematical foundations that we have developed.

This understanding allowed us to develop the Weighted Sum of Rician's (WSoR) scaling method for noise unbiased multivariate analysis. On reflection, we agree with the reviewer that we did not clearly present and compare the performance with other scaling methods. We have now extended the analysis of the mouse testis data as a main figure. We have also included an additional biological imaging data set of rat liver using desorption electrospray ionisation (DESI) Orbitrap MS (A. Dannhorn et al, Anal. Chem. 2020, 92, 16, 11080–11088). To make the effects of noise bias clearer for each biological example, we show how the variance for no-scaling, variance, Pareto, RMS and Probabilistic Factor Analysis (PFA) used for the covariance matrix for scaling compares to the fundamental WSoR model. In every case, the WSoR model strongly agrees with the machine learned variance using probabilistic factor analysis (PFA) validating our method using 3 biological data sets.

These plots highlight how the different models introduce bias by down-weighting or upweighting peaks. The effects of bias on the multivariate scores images are now clearer to understand. For example, in the drosophila CNS example, there are 8 chemical components that are significant with respect to the noise level. Variance scaling performs poorly down-weighting intense ions so that chemical components are found in high principal components amongst noise. In the testis data example, the intensities are lower and Pareto scaling now performs poorly, down-weighting most ions and features found in component 4 for WSoR scaling are not found until component 34. The DESI data set demonstrates the generality of our method to a completely different ionisation source as well as independently generated data. In this example, eigenvalue analysis finds 7 distinct components. Variance, Pareto and root-mean scaling all introduce a bias that leads to chemical components being relegated to high principal components defeating the purpose of multivariate analysis.

In each example, we find that the WSoR provides noise unbiased scaling allowing the chemical information to be captured efficiently in the leading Principal Components and separated from the noise. In contrast, no-scaling, variance scaling, Pareto scaling and RMS are sometimes good or poor on a case-by-case basis depending on the distribution of ion intensities. This leads to inefficient capture of the chemical components with some mixed between noise components.

We hope that our revisions more clearly show the biases introduced by the commonly used scaling method adopted by the community with a rationale and how the efficiency of multivariate analysis is affected by them.

Minor comments:

Page 16, line 429: Please list the papers, also I assume the most cited publications are older publications, maybe more recent studies contain more information on data scaling.

Thank you, we now provide this in Supplementary Table 2.

Page 20, line 508: Rephrase, do you mean: all scaling and log transformation are effective...

Line 513: "Pareto scaling... ..may be less robust for complex, biological OrbiSIMS imaging datasets." Is this assumption supported by your experimental?

Log-transformation is always very poor. We have substantially revised the text and figure and hope that this is now clearer.

Figure 5j: The figure is too small to be useful, add labels for the highest loadings or add a bigger version to the supplementary information, the loadings in the SI are too small as well.

Thank you, we have revised the figure.

Line 518 ff: The references to the Figure 5 b-h are mixed up in the entire paragraph.

We have substantially revised the text and figure to correct this.

Page 20 line 540: "ether-linked Pes" change to PEs

As part of the major revisions this text has been deleted.

Reviewer #2

Keenan et al. propose a theoretical model of the noise generated by the Orbitrap mass analyzers in the low intensity region. For modeling, they use data generated by SIMS which provides a consistent

input ion current with known noise properties. Moreover, they propose a new method (WSoR) for data rescaling that capitalizes on the knowledge of the noise model.

The manuscript strengths are in the theoretical work to mathematically model the noise. The manuscript weaknesses are in the validation and in demonstrating the practical impact of the proposed theory.

We are very grateful to Reviewers 2 and 3 for their positive comments on the mathematical model and we hope that the additional work that we have done including more analysis on the testis data and analysis of an additional DESI Orbitrap data demonstrates more clearly the practical impact of our method. We provide further details on these additions in the response to Reviewer 1.

A key concern is about the validation of the proposed WSoR method. The authors do it in three ways: through data simulation (Figure 4), through discovery of the Cs⁺ contamination in first N principal components, and through application to an imaging dataset. However, the simulation is still a theoretical example and the practical use and impact for discovering the Cs⁺ contamination is not clear.

To address the reviewers' concern we have removed the simulations and have clarified the use of the silver data as a simple data set acting as a control. We have now developed a method to clearly show the bias that is introduced for principal component analysis with different scaling methods and if no scaling is used and validate this using three biological data sets. For further details, please see our response to Reviewer 1

Re application to an imaging dataset, Figure 5i shows no improvement for WSoR compared to "no scaling" in terms of discovering spatial patterns. For each spatial pattern (what the authors call "CNS pattern") shown in PCA scores for WSoR-transformed data, there is a similar spatial pattern visible in the PCA scores in "no scaling" results.

The no-scaling data set is poor at capturing the information in the leading principal components and this is demonstrated by noise beginning to dominate the scores images beyond the 7th PC. The WSoR method is optimally parsimonious containing the biological information with the first 8 PCs. To demonstrate the difference we show in Reviewer's Figure R1 the correlation between no-scaling and the 8 component WSoR model, analogous to new Fig 4b for variance scaling. This shows that with unscaled PCA over 50 components would be required to describe the data with the same fidelity as the 8 component WSoR model. This is because chemical information is relegated to higher PCs with no-scaling.

Reviewer’s Figure R1 Correlation between no-scaling with the 8-component WSoR model for the CNS data of Figure 4.

2. It’s not quite clear what’s the basis of the molecular interpretations in Figure 5 because of two factors. First, there are no details on how the molecular identification was done. The authors should briefly comment on the methodology used for the reported molecular assignments. If they used the accurate m/z matching, they should share the details on the database (if any), the m/z tolerance used, and the m/z delta between the theoretical m/z and the observed one.

We thank the reviewer for pointing out this omission in our methods section. We have added a sentence describing that molecular assignments were performed by accurate matching using the CEU mass tool (Gil-de-la-Fuente *et al.*, 2019), which compares data against the HMDB, LipidMaps, Metlin, Kegg and in-house CEU mass libraries. The m/z tolerance was set at 2 ppm during ID searching however, as shown in the table below, all the putative IDs we have made have a mass accuracy <1 ppm. We have added this information into the supplementary data.

m/z	Formula (experimental)	Putative I.D.	Adduct	Mass deviation (ppm)
134.047	C ₅ H ₄ N ₅ ⁻	Adenine	[M-H] ⁻	0.2
150.042	C ₅ H ₄ N ₅ O ⁻	Guanine	[M-H] ⁻	-0.2
111.020	C ₄ H ₃ N ₂ O ₂ ⁻	Uracil	[M-H] ⁻	-0.1
255.233	C ₁₆ H ₃₁ O ₂ ⁻	C16:0	[M-H] ⁻	-0.3
253.217	C ₁₆ H ₃₁ O ₂ ⁻	C16:1	[M-H] ⁻	-0.3
277.217	C ₁₈ H ₂₉ O ₂ ⁻	C18:3	[M-H] ⁻	-0.7
279.233	C ₁₈ H ₃₁ O ₂ ⁻	C18:2	[M-H] ⁻	-0.7
281.248	C ₁₈ H ₃₃ O ₂ ⁻	C18:1	[M-H] ⁻	-0.6
283.264	C ₁₈ H ₃₅ O ₂ ⁻	C18:0	[M-H] ⁻	-0.6
657.498	C ₃₆ H ₇₀ N ₂ O ₆ P ⁻	PE-Cer(34:2)	[M-H] ⁻	0.4
702.545	C ₃₉ H ₇₇ NO ₇ P ⁻	PE(O-34:1)	[M-H] ⁻	0.4

It may additionally be worth mentioning that annotations of plasmalogen lipids and nucleotides are commonly the result of fragmentation reactions rather than intact detection. Moreover, the reported PE-Cer(34:2) is isobaric to PA(34:1) which can be a fragment of larger lipids.

We appreciate that this interpretation may not have been so clear in our original text, so we have clarified that the nucleobases we have putatively identified as (adenine, guanine and uracil) are detected as a result of fragmentation of DNA/RNA by adding an additional sentence describing this to line 524 on page 23.

We have putatively identified the plasmalogen lipids as [M-H]⁻ ions, as in our experience, lipids ionized with the OrbiSIMS argon GCIB form primarily intact molecular ions, with additional fragments that originate from loss of the entire headgroup or entire fatty acid chains (as opposed to the high degree of fragmentation induced by the bismuth LMIG) (Passarelli et al, 2017; Newell et al, 2020). Therefore, these fragments do not resemble other large lipid classes.

PE-Cer(34:2) and PA(34:1) are not isobaric as [M-H]⁻ ions, which is how the majority of intact lipids ionize with the OrbiSIMS argon GCIB. PA(34:1) has a very different m/z value ($C_{37}H_{71}O_8P$, m/z 673.481) compared to PE-Cer(34:2) ($C_{36}H_{71}N_2O_6P$, m/z 657.498). The reviewer may be referring to PA(O-34:2) which is the second search result on LIPID MAPS after PE-Cer(34:2). However, the mass deviation for PE-Cer(34:2) is 0.416 ppm whereas PA(O-34:2) is 17.491 ppm. The instrument calibration result on the day of image acquisition (with silver) at m/z 538.52534 was 0.105 ppm and at m/z 754.33519 was 0.114 ppm. Therefore, we are confident that this ion is not PA(O-34:2) due to the high mass deviation for this assignment.

Second, the claimed correspondence between the mass spectrometry signals and cell types is not supported with any evidence.

The claims about mass spectrometry signals and cell types are stated as hypotheses based on existing literature, and the location of the cell types referenced is well established in the literature which has been cited accordingly. The *Drosophila* larval CNS has a very distinct and reproducible pattern of cell-type distribution (we have updated the anatomical schematic in Figure 5a and shown below), which makes it an ideal test-application for this method.

3. A brief inspection of the raw data behind Figure 5 raises some questions about the interpretation (lines 501-553), especially the fatty acid analysis (subfigure 5b). Looking at individual images and intensity ratios, the statement about polyunsaturated vs saturated fatty acids does not appear to hold up. The pattern shown in 5b is caused mainly by the fact that palmitic acid (16:0) has a very different distribution from the other peaks. The inclusion of C14 fatty acids is somewhat redundant as only 14:0 and 14:1 are present, and a full order of magnitude below the corresponding C16 and C18 signals. Similarly, 16:2 does not contribute meaningfully to the sum, meaning that the image is indistinguishable from 18:2 / (18:0+16:0). In this set, I would argue that 18:2 and 18:0 are strongly correlated, which is the opposite observation from what the authors discuss! This also is readily apparent from a quick manual inspection, and does not really provide a good example of multivariate analysis used for hypothesis generation.

Thank you for these important points. In our images, all the major FAs we have examined (regardless of whether they are polyunsaturated or saturated) gave a higher signal in the neuropil compared to the cortex (Reviewer's Figure R2). This also applies to 16:0 although as the reviewer suggests the signal does appear somewhat higher in the cortex than with other FAs. In line with these results, the ratios of polyunsaturated (C18:2, C16:2) to saturated (C18:0, C16:0) FAs are also higher in the neuropil than the cortex.

Reviewer's Figure R2: Individual images of major FAs analysed in this study.

The top three chain-lengths detected in the *Drosophila* CNS are from highest to lowest signal: C18, C16 and C14 fatty acids. C14 FAs are still considerably above background level thus they are included. Nevertheless, in line with Reviewer's request, we have repeated the analysis of PUFA:SFA ratios without including C14 FAs (Reviewer's Figure R3). This analysis shows a broadly similar results to the original PUFA:SFA ratios that included C14 FAs.

Reviewer's Figure R3: Ratio images with and without C14 FAs.

We agree entirely with the Reviewer that the ratios shown are readily apparent from manual inspection of individual images. This is a good thing because it provides a proof-of-principle sanity check on our automated analysis. Moving forward, manual inspection for many large datasets would be very time-consuming and challenging to identify patterns. The algorithm provided in this paper offers an unbiased and automated method for detecting patterns from mass spectrometry imaging, with significant advantages for analysing large image datasets.

4. Within the imaging dataset analysis (lines 501-553), figure panels are referred to with the wrong figure number, e.g. references to 5b as 5d on line 519; 5e as 5g on line 530; 5g as 5h on line 539. The data in 5h (ratio of C18 to C16 fatty acids) is not referred to anywhere.

Thank you, we have completely revised the text and figures and this is now corrected.

5. The testis analysis in supplementary note 9 is not mentioned in the main text, but is listed as the main individual contribution of two of the co-authors. The authors may want to consider including a short statement about that analysis and the value it adds to the manuscript.

We thank the reviewer for this suggestion. We have done additional analysis on the testis data and added a new main Figure. This data set is an interesting one as the signal intensities are low providing a good complement to the CNS and DESI data sets. In this case, no scaling and Pareto scaling perform poorly since low intensity ions are down-weighted. With WSoR scaling, two localisations relating to $C_4H_8N_3O_2$ are discovered in PC4, which are not discovered until PC39 with no scaling and PC34 with Pareto scaling. Further details on the variable performance of scaling methods are given in the response to Reviewer 1.

Reviewer #2 (Remarks on code availability):

There are no installation instructions provided and no README file
Details have now been included in the program header.

Reviewer #3

We thank the reviewer for their comments which have helped us improve the clarity of the manuscript.

Orbitrap noise structure and method for noise unbiased multivariate analysis

We are very grateful for the reviewers' comments. In the following we provide a point-by-point response to the reviewers.

Reviewer #1

I appreciate the extensive responses to my comments and the additional experiments. My concerns have been addressed and the significance of the model for biological data-sets is way clearer now. In my opinion, no additional experiments are needed.

We are glad the revised version addressed all of the reviewer's concerns.

I only have a few minor comments:

Figure 5a WSoR PC6 seems to highlight a feature captured by no other scaling method. Is this anything of interest?

We thank the reviewer for pointing this out. We have looked into the details and WSoR PC6 is describing an artefact of the Orbitrap, which is important information for studies using isotopic labelling strategies. It results from the Orbitrap software censoring data below a threshold. Isotopologues, or other strongly correlated peaks, should all scale together linearly however those with very low intensity, for example the $^{13}\text{C}_2\text{C}_{39}\text{H}_{79}\text{O}_{12}\text{S}^-$ isotopologue, fall below the threshold in some pixels and are censored, giving rise to a non-linear effect. We have added a sentence to the main text to describe this. This artefact as well as Orbitrap linearity will be discussed in detail in a future paper.

Line 572: why not test on the full data-set? Is it too computationally demanding? The 2 other bio data-sets use 100-500 peaks.

The DESI data set is very large and complex. We selected a sub-set of peaks with distinct spatial distributions to that provide a more visual example of the differences between no-scaling and the different scaling methods.

Line 573ff: What do mean with group C/D, this is not explained anywhere in the text or SI. Is there a group A/B? This section needs to be slightly reworked.

We thank the reviewer for pointing out this source of confusion. In fact, there are only groups C and D, where C stands for “clustered” and D for “distributed”. To avoid this confusion we have now renamed those groups A and B in the text and Figure.

Reviewer #2 (Re-addressing of comment from previous round of revisions)

3. A brief inspection of the raw data behind Figure 5 raises some questions about the interpretation (lines 501-553), especially the fatty acid analysis (subfigure 5b). Looking at individual images and intensity ratios, the statement about polyunsaturated vs saturated fatty acids does not appear to hold up. The pattern shown in 5b is caused mainly by the fact that palmitic acid (16:0) has a very different distribution from the other peaks. The inclusion of C14 fatty acids is somewhat redundant as only 14:0 and 14:1 are present, and a full order of magnitude below the corresponding C16 and C18 signals. Similarly, 16:2 does not contribute meaningfully to the sum, meaning that the image is indistinguishable from 18:2 / (18:0+16:0). In this set, I would argue that 18:2 and 18:0 are strongly correlated, which is the opposite observation from what the authors discuss! This also is readily apparent from a quick manual inspection, and does not really provide a good example of multivariate analysis used for hypothesis generation.

We agree with the reviewer that a ratio of the sums is not appropriate owing to the low intensities for some of the ions and our previous general statement is not supported. The reviewer’s comment stimulated us to investigate the spatial distribution of unsaturated and saturated fatty acids in more detail. PCA (with WSoR scaling) of a subset of data containing ten C16 and C18 fatty acid peaks (**Reviewer’s Figure R1**) shows that fatty acids are predominantly in the neuropil (PC 1) but that the distribution is nuanced (PC 2). From PC 2, we see there is no correlation of C16:0 and C18:0 or C18:1 and C16:1. However, we find that cortex/neuropil variation does reflect differences in saturation with the strongest anticorrelation between C16:0 and C18:1. Comparison of the PC2 loading with WSoR PC7 in the paper shows they are nearly the same. In other words, the subtle variation in saturation is captured by WSoR in a PC that is not found without scaling, highlighting the effect of bias if appropriate scaling is not used.

In the revision we have removed the general statement about the spatial distribution of saturated and unsaturated fatty acids and have replaced old Fig 5f with the specific example for C16 and C18 (new Fig 4f,g). We have also replaced the previous text with the following (lines 520-524) *“WSoR PC 7 (Figure 4a,d) reveals that C16:0 and C18:1 are anticorrelated with different spatial distributions. We find that the signal intensity ratio of saturated to unsaturated fatty acids C16:1 to C16:0 (Figure 4f) is enhanced in the neuropil whereas the ratio C18:1 to C18:0 (Figure 4g) is stronger at the periphery of the cortex along the blood-brain barrier.”* We hope that this is now clearer and thank the reviewer for pointing out the error.

Reviewer's Figure R1: Principal component analysis (with WSoR scaling) of a subset of the CNS dataset containing only 10 fatty acid peaks.